# The selfish yeast plasmid exploits a SWI/SNF-type chromatin remodeling complex for hitchhiking on chromosomes and ensuring high-fidelity propagation

Chien-Hui Ma[1]☯, Deepanshu Kumar[2]☯, Makkuni Jayaram[1], Santanu K. Ghosh[2]*, Vishwanath R. Iyer[1,3]*

1 Department of Molecular Biosciences, The University of Texas at Austin, Austin, Texas, United States of America, 2 Department of Biosciences and Bioengineering, Indian Institute of Technology Bombay, Powai, Mumbai, India, 3 Livestrong Cancer Institutes and Department of Oncology, Dell Medical School, The University of Texas at Austin, Austin, Texas, United States of America

☯ These authors contributed equally to this work.
* santanughosh@iitb.ac.in (SKG); vishy@utexas.edu (VRI)

**Data Availability Statement:** The ChIP-seq and CUT&Tag-seq data generated during this study are available in GEO under the accession number

## Abstract

Extra-chromosomal selfish DNA elements can evade the risk of being lost at every generation by behaving as chromosome appendages, thereby ensuring high fidelity segregation and stable persistence in host cell populations. The yeast 2-micron plasmid and episomes of the mammalian gammaherpes and papilloma viruses that tether to chromosomes and segregate by hitchhiking on them exemplify this strategy. We document for the first time the utilization of a SWI/SNF-type chromatin remodeling complex as a conduit for chromosome association by a selfish element. One principal mechanism for chromosome tethering by the 2-micron plasmid is the bridging interaction of the plasmid partitioning proteins (Rep1 and Rep2) with the yeast RSC2 complex and the plasmid partitioning locus *STB*. We substantiate this model by multiple lines of evidence derived from genomics, cell biology and interaction analyses. We describe a Rep-*STB* bypass system in which a plasmid engineered to non-covalently associate with the RSC complex mimics segregation by chromosome hitchhiking. Given the ubiquitous prevalence of SWI/SNF family chromatin remodeling complexes among eukaryotes, it is likely that the 2-micron plasmid paradigm or analogous ones will be encountered among other eukaryotic selfish elements.

## Author summary

The yeast 2-micron plasmid, a selfish DNA element, propagates stably without benefiting or harming its host. The plasmid strategy of chromosome-hitchhiking for long-term persistence is shared by mammalian papilloma and gammaherpes viruses despite ~1.5 billion years of evolutionary divergence between their respective hosts. Viral episomes tether to chromosomes by the interaction of viral partitioning proteins with chromosomes directly, with histones or with chromatin-associated host proteins. The mechanism by which the

GSE225582 (https://www.ncbi.nlm.nih.gov/geo/query/acc.cgi?acc=GSE225582). The mass spectrometry data have been deposited to the ProteomeXchange Consortium via the PRIDE [98] partner repository with the dataset identifier PXD044470. The numerical data underlying the plots shown in Figs 10–12 and S12–S15 are available at https://www.bio.iitb.ac.in/skglab/PGENETICS-D-23-00727R1_numerical_data.xlsx.

**Funding:** This work was supported by NSF grant MCB1949821 to MJ and VRI and a research grant from Department of Science and Technology, India (SERB-CRG/2020/000444) to SKG. CM and VRI were supported in part by MCB1949821. DK was supported by a fellowship grant from Department of Biotechnology, India (DBT/2018/IIT-B/1059). The funders had no role in study design, data collection and analysis, decision to publish, or preparation of the manuscript.

**Competing interests:** The authors have declared that no competing interests exist.

plasmid partitioning proteins Rep1 and Rep2 bound at the partitioning locus *STB* localizes the plasmid on chromosomes has remained elusive. Using genomics, interaction analyses and cell biology, we demonstrate Rep1-Rep2 interaction with the yeast RSC2 chromatin remodeling complex as the molecular basis for plasmid-chromosome bridging. We validate this model by showing that an engineered system for direct plasmid-RSC2 tethering, bypassing Rep1-Rep2, recapitulates the three hallmarks of native 2-micron plasmid propagation, namely, chromosome association, equal segregation and escape from mother bias.

## Introduction

Selfish genetic elements, ubiquitous among prokaryotes and eukaryotes, include plasmids, viruses, DNA transposons, retrotransposons, repetitive sequence families and non-coding RNAs, among others [1,2]. They exploit the host's genetic endowments and/or embody mechanisms for efficient replication and segregation to ensure their stable propagation. Their genetic load and the associated fitness cost to the host are often optimized by regulatory schemes implemented by the elements themselves and/or by their hosts. The multi-copy 2-micron plasmid of *Saccharomyces* yeast epitomizes an extrachromosomal selfish genome whose organization and functions are tailored for long-term stability and copy number maintenance, while minimizing fitness conflicts with the host genome [3–5]. Here we examine how the plasmid takes advantage of chromosome segregation for its equal partitioning during cell division without jeopardizing the fidelity of chromosome segregation.

Autonomously replicating yeast nuclear plasmids (*ARS*-plasmids) lacking a centromere (*CEN*) are rapidly lost from the host cell population. The constricted nuclear geometry at the bud neck and the relatively short duration of the cell cycle cause the vast majority of the plasmid molecules to be trapped in the mother nucleus during cell division [6]. This mother bias is strong (~90%) for standard *ARS*-plasmids [7]. In sharp contrast, the 2-micron plasmid, native to yeast, is almost as stable as the chromosomes of its host. Following duplication of individual plasmid molecules by the host replication machinery [8], their equal or nearly equal segregation is mediated by a partitioning system comprised of two plasmid-coded proteins (Rep1 and Rep2) and a *cis*-acting locus *STB* to which these proteins are recruited [3,5]. Furthermore, the Flp recombination system harbored by the plasmid protects against a potential reduction in copy number due to rare missegregation events [9,10]. Amplification is triggered by a replication coupled recombination event that reorients the normal bi-directional forks to uni-directional forks, initiating rolling circle-like replication. Transcriptional regulation instituted by the plasmid and post-translational regulation imposed by the host ensure a quick amplification response when needed without the danger of a runaway increase in plasmid copy number [11–16]. The Rep-*STB* partitioning system, augmented by the Flp amplification system, is responsible for the evolutionary success of the 2-micron plasmid as a selfish extra-chromosomal element.

How does the Rep-*STB* system overcome the mother bias in plasmid segregation? In principle, equipartitioning of a nuclear entity between the mother and daughter nuclei is possible by utilizing spindle-mediated active movement or the fluidity and dynamics of the nuclear membrane [17–19]. The spindle pole body (SPB), chromosomes and an *ARS*-plasmid harboring a *CEN* exemplify the first mechanism, while nuclear pores and an *ARS*-plasmid tethered to nuclear pores typify the second mechanism. Current evidence precludes kinetochore assembly at the *STB* locus and direct attachment of the 2-micron plasmid to the mitotic spindle [20]. Membrane associated segregation of the plasmid is unlikely based on its nuclear localization in

mitotic cells [21,22]. Furthermore, artificial tethering to the nuclear membrane, which increases the mitotic stability of an *ARS*-plasmid [18], has the opposite effect on an *STB*-plasmid [23].

According to the current model, supported by a confluence of experimental evidence, the 2-micron plasmid escapes mother bias by taking advantage of spindle force indirectly. The plasmid physically associates with chromosomes assisted by the Rep-*STB* system, and segregates by hitchhiking on them [5,23,24]. There is precedent for the propagation of selfish DNA elements by chromosome-hitchhiking, as exemplified by the episomes of mammalian gamma-herpes and papilloma viruses [25–32]. The ~60 plasmid molecules (~120 after replication) per haploid yeast nucleus are organized into three to six clustered foci, and each focus appears to be a segregation unit. Single-copy *STB-r*eporter plasmids indicate two possible types of plasmid segregation, one replication-dependent and the other replication-independent [23]. While chromosome-hitchhiking is a common feature of both, replication is a pre-requisite for chromosome-like one-to-one plasmid segregation. It is believed that host factors assembled at *STB*, in particular the cohesin complex recruited during replication, promotes a directed form of chromosome hitchhiking, namely, hitchhiking of plasmid sisters on sister chromatids [23,24,33]. By extension, it is proposed that sister clusters hitchhike on sister chromatids in the native multi-copy plasmid system.

In the present study, we address the molecular mechanisms of plasmid-chromosome association that precede hitchhiking. We demonstrate that the Rep-*STB* system exploits a SWI/SNF-type chromatin remodeling complex (RSC) to bring about plasmid-chromosome tethering. Of the two yeast RSC complexes RSC1 and RSC2 [34], apparently redundant to a considerable extent in host physiology, only RSC2 is functionally relevant to tethering. Our evidence includes extensive overlap of genome-wide ChIP-seq and CUT&Tag peaks of Rep1-Rep2 with those of Rsc2 and other RSC components, genetic and physical interactions of Rep1-Rep2 with the RSC2 complex, frequent colocalization of Rep1-Rep2 and an *STB*-plasmid with Rsc2 in chromosome spreads, and the proximity of a single-copy *STB*-reporter plasmid to tRNA loci, one of the highly preferred targets of RSC2 association. Furthermore, non-covalent tethering of an *ARS*-plasmid to an RSC component (Sfh1) results in plasmid-chromosome association and increases equal plasmid segregation, while also reducing mother bias during missegregation. Beyond corroborating earlier studies suggesting a role for the RSC2 complex in 2-micron plasmid segregation [35,36], the present findings offer a mechanistic interpretation for this role. They reveal for the first time how an extra-chromosomal selfish genome utilizes a chromatin remodeling complex as a conduit to gain access to chromosomes and to achieve chromosome-like fidelity in segregation.

## Results

### Rep1 and Rep2 are localized to specific regions of yeast chromosomes

Our previous fluorescence microscopy-based analyses revealed frequent localization of the 2-micron plasmid at or near *CEN*s and telomeres (*TEL*s), with a bias towards the latter [5,22,37]. We suspected that the molecular basis of this localization is the preferred occupancy of certain chromosome sites by the Rep1-Rep2 proteins together with the associated plasmid, presumably by their interaction with chromatin bound protein(s). As a first test of this hypothesis, we used ChIP-seq to determine the chromosomal locations of Rep1 and Rep2. We expressed epitope-tagged Rep1 and Rep2 under the control of the inducible *GAL1-10* promoter in strains cured of the endogenous 2-micron plasmid ([Cir$^0$]). Potential titration and interference by the Rep proteins expressed from this high copy number plasmid in [Cir$^+$] strains are thus avoided.

Rep1 and Rep2 showed concordant association with hundreds of distinct chromosomal locales compared to negative controls (either a strain lacking the epitope tag or expressing from the *GAL* promoter a mutant allele of the Flp recombinase that binds its target DNA site (*FRT*) but is inactive in recombination). These locales included *CEN*s and *TEL*s as well as tRNA and snoRNA genes (S1A and S1B Fig). It is notable that *TEL*s and *CEN*s are typically silenced or transcriptionally quiescent chromatin whereas tRNA and snoRNA loci are highly transcribed chromatin.

As transcriptionally active regions are susceptible to showing artifactual ChIP-seq enrichment signals [38,39], we used CUT&Tag, which does not entail chromatin crosslinking and immunoprecipitation [40], as an independent genomic method to verify the localization of Rep1 and Rep2. Using this alternative approach, Rep1 and Rep2 again showed clear localization at individual tRNA and snoRNA loci (Fig 1A), as well as at *CEN*s and *TEL*s (Fig 1B). Interestingly, the peaks of CUT&Tag were occasionally offset from the ChIP-seq peaks, typically at *CEN*s, indicative of differences in the ends of the tagged fragments that were recovered in CUT&Tag relative to ChIP-seq at some classes of sites (Fig 1B). Heat maps across all *CEN*s and *TEL*s were consistent with Rep1-Rep2 localization at these loci, with stronger signals for Rep1 than Rep2 (Fig 2A and 2B). Furthermore, Rep1 was detected at essentially all centromeres, and at the majority of, though not all, telomeres.

We noted some differences with regard to how the Flp negative control behaved in the ChIP-seq and the CUT&Tag assays. At snoRNA and tRNA genes, the Flp control tended to show low backgrounds in both assays relative to the Rep1 and Rep2 signals (Fig 1A). At *TEL*s and *CEN*s on the other hand, the Flp control showed noticeable signal, comparable to that of Rep2, in CUT&Tag, while it was quite low in the ChIP-seq assay (Fig 1B). We don't currently understand the reasons for these differences. At *TEL*s, the no antibody control in the CUT&Tag assay showed apparently higher average signal than Rep1; however, this is caused by strong background signal at a small number of *TEL*s. At most *TEL*s where there was any appreciable Rep1 signal, it was higher than the controls (Fig 2A). The smaller peaks for Rep2 at *CEN*s and *TEL*s than those for Rep1 (Figs 1B and 2) suggest that, in the chromatin bound Rep1-Rep2 at these loci, the epitope tag on Rep2 may be partially obscured to the antibody probe. Previous studies demonstrated that Rep1 and Rep2 associate with each other strongly and are colocalized in yeast nuclei or in yeast chromosome spreads [23,41–43].

Because the epitope-tagged Rep proteins were expressed under the control of the *GAL1-10* promoter in these assays, we compared their expression under galactose induction with their native expression levels. The galactose-induced steady state level of Rep2 was comparable to that observed for its expression from the multi-copy 2-micron plasmid promoter (S2A Fig). For Rep1, the expression from the native promoter was stronger (S2A Fig) than, or similar to, that from the *GAL* promoter. Moreover, the occupancy of the *tV(UAC)D* (tRNA valine) locus (see Fig 1A) was comparable between the two sets of strains, indicating that galactose-induced overexpression alone was not responsible for the ChIP signal (S2B Fig).

Taken together, the consistently high signals for Rep1 and Rep2 in two independent assays support the frequent localization of the plasmid partitioning proteins at distinct classes of loci in the yeast genome. As implied in our hypothesis, this localization could underlie the *STB*-mediated recruitment of the 2-micron plasmid to these hitchhiking sites for plasmid segregation. Given the relatively small number of clustered plasmid foci (three to six) present in the nucleus, only a subset of the hitchhiking sites will be occupied by the plasmid in any given cell.

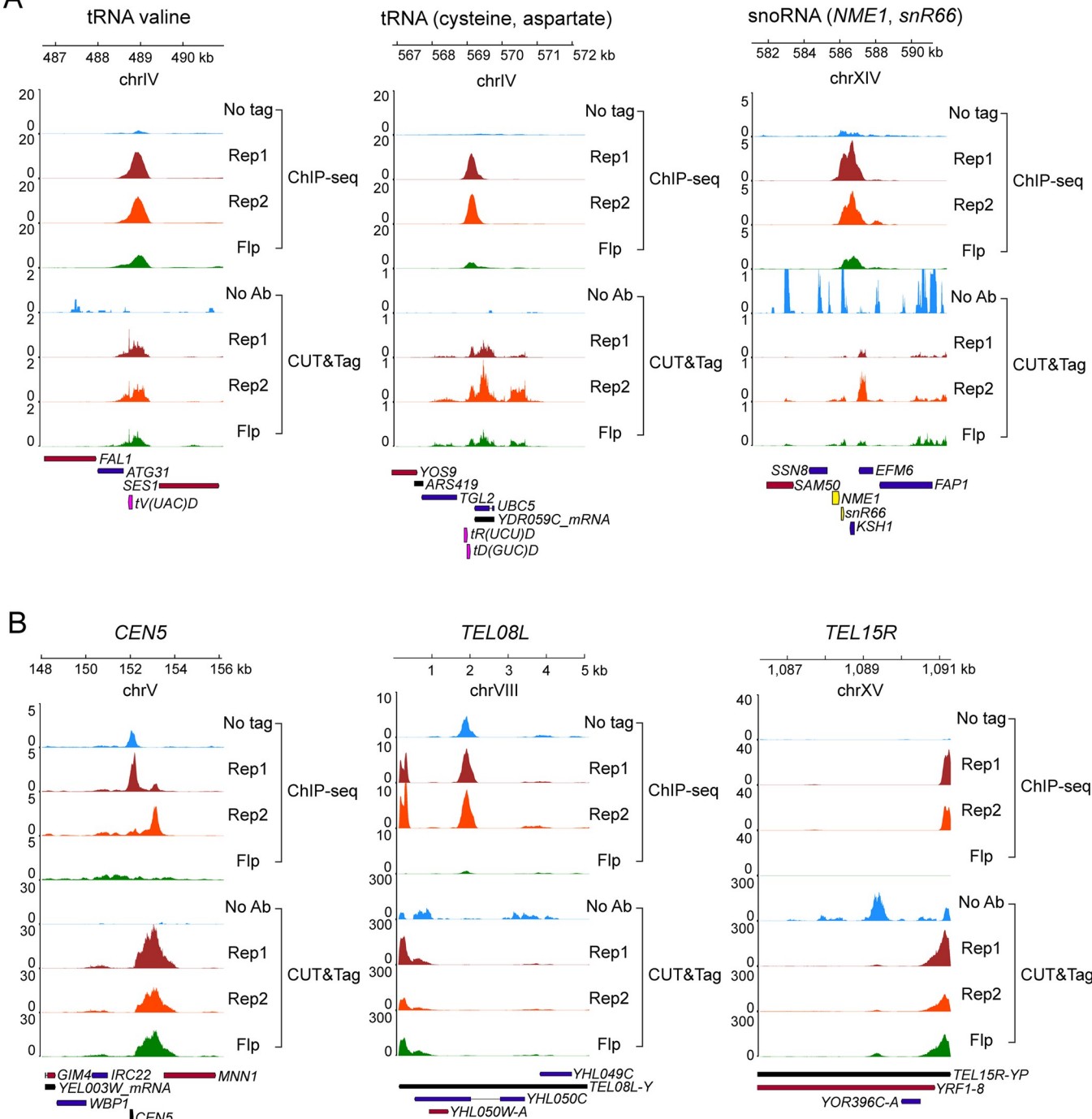

**Fig 1. Rep protein localization at characteristic genomic loci.** Binding of Rep1 and Rep2 assayed by ChIP-seq or CUT&Tag is shown using signal tracks for each region along with negative controls. For ChIP-seq, the signal tracks are input-corrected, and the negative controls are 'No tag' and 'Flp'. For CUT&Tag, there is no input sample so signal tracks are not input-corrected, and the negative control shown is the Rep1-Myc epitope tagged strain processed without an antibody ('No Ab'). The Rep2- and Flp-Myc strains processed without an antibody were similar as controls. **(A)** Transcriptionally active regions (tRNA and snoRNA loci) including the tRNA valine locus used in colocalization assays in Fig 11. **(B)** Transcriptionally quiescent loci including *CEN5* and *TELs*.

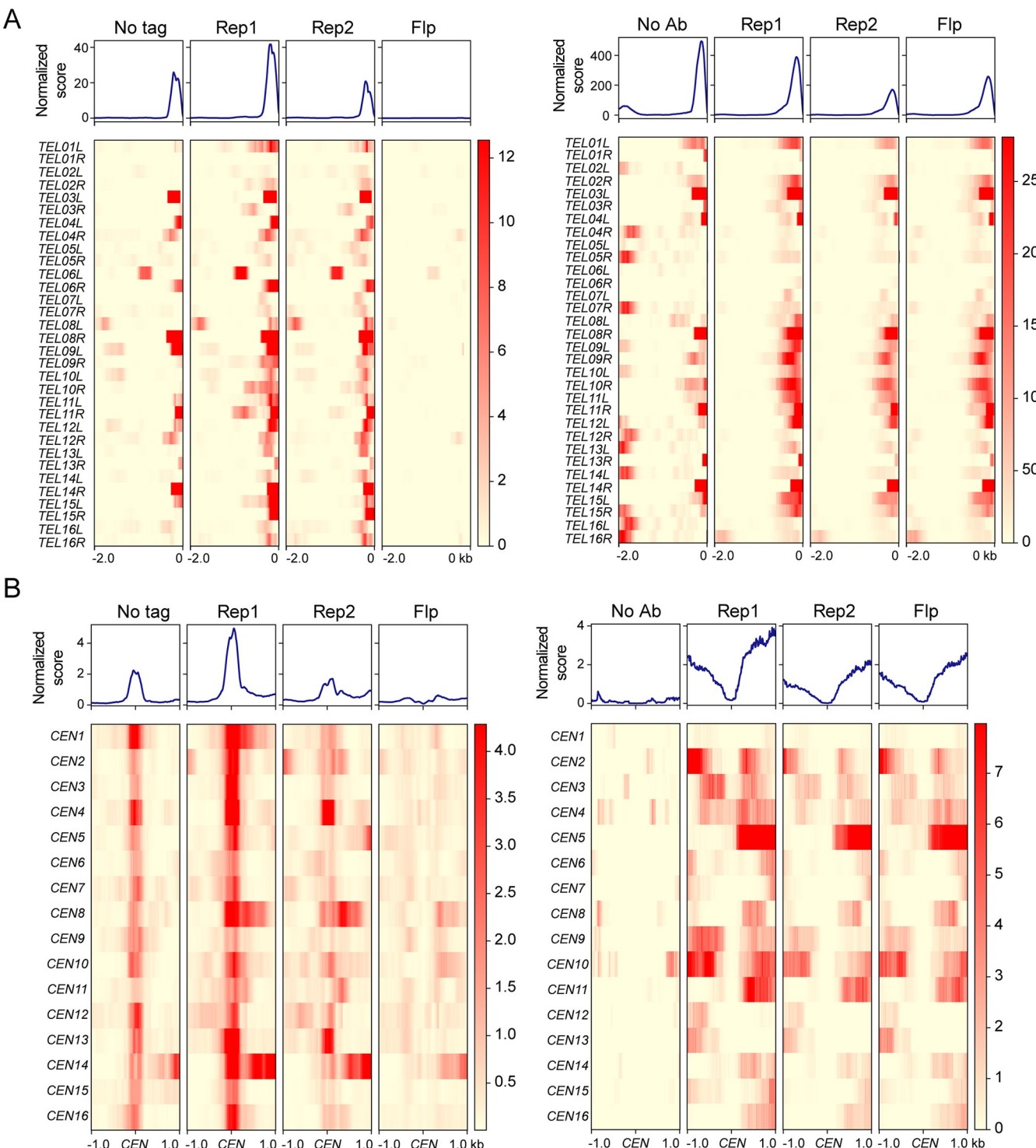

**Fig 2. Rep protein localization at TELs and CENs.** The top part of each panel shows the average binding profile and the bottom shows binding to each region as a heat map. The average binding profile is the normalized enrichment score calculated by MACS2 for ChIP-seq and for CUT&Tag. (A) *TELs*, with ChIP-seq data on the left and CUT&Tag on the right. *TELs* are all oriented so that the ends are on the right, with a 2 kb region shown to the left. (B) *CENs*, with ChIP-seq data on the left and CUT&Tag on the right. Profiles and heat maps show a 2 kb window centered on the *CEN*.

## Rep protein occupancy correlates with that of RSC2

Our previous studies suggested that Rep1 and Rep2 interact with members of the yeast RSC chromatin remodeling complex [35]. We therefore examined whether the association between Rep proteins and the RSC complex is reflected in their genomic localization patterns, using occupancy data for RSC that were previously generated by us [38,44].

Indeed, there was a strong correlation between the occupancy patterns of Rep1/Rep2 and Rsc2, a component of the yeast RSC complex, when compared to other DNA-associated factors and chromatin remodelers (Fig 3A and 3B). Similar to Rep1 and Rep2, Rsc2 was also localized strongly to tRNA loci, supporting the co-localization of the plasmid partitioning proteins and the RSC complex at specific chromosomal locales (S3 Fig).

To extend this observation, we examined the occupancy of additional members of the RSC complex using other previously published datasets [45–49]. We found that Rep1 and Rep2 occupancy was correlated with that of Sth1, Sfh1 and Rsc8 in addition to Rsc2, all components of the RSC complex. The occupancy of these additional RSC components was similar to that of Rep1 and Rep2 around individual tRNA, snoRNA and *CEN* loci (Fig 3C). Genome-wide, the correlation of Rep protein occupancy with that of the RSC components as measured in various datasets was as good as, or higher than, the correlation among RSC component occupancies measured in independent published studies and higher than the correlation between Rep1 or Rep2 and unrelated chromatin factors (Fig 3D).

Our present results, in concert with previous findings, suggest that Rep1-Rep2 interaction with the RSC complex is at least one mechanism by which the Rep proteins and, by inference, the 2-micron plasmid, associate with chromosomes. Furthermore, the plasmid appears to exploit RSC specifically among the yeast chromatin remodeling complexes for chromosome hitchhiking.

## RSC2 is required for normal genomic localization of Rep1

Previous genetic analysis indicated that Rsc2 and other components of the yeast RSC complex are required for stable maintenance of the 2-micron plasmid in dividing yeast cells [35,36]. The two distinct RSC complexes of yeast differ from each other in a single constituent protein —Rsc1 in the RSC1 complex and Rsc2 in the RSC2 complex [34]. Rsc1 and Rsc2 are highly homologous proteins containing multiple bromodomains, a bromo-adjacent homology (BAH) domain and a DNA binding AT-hook/HMG-box motif [50], yet *STB*-plasmid instability is increased only by *rsc2Δ* or an *RSC2*-truncation and not by *rsc1Δ* [36]. Thus, the 2-micron plasmid appears to rely specifically on the RSC2 complex for hitchhiking on chromosomes.

In order to test whether the integrity of the RSC2 complex is required for the normal chromosome localization of Rep1 and/or Rep2, we repeated the ChIP-seq and CUT&Tag experiments in two yeast strains mutant for the Rsc2 protein. In one, we deleted the entire *RSC2* coding sequence (*rsc2Δ*). In the other, we introduced a previously described frameshift mutation in *RSC2* which effectively truncates the C-terminal two-thirds of the protein (truncated *rsc2*) [36]. Consistent with previously reported impairment of 2-micron plasmid maintenance by *rsc2Δ* and *rsc2* truncation [36], plasmid loss rate per generation was increased in both our mutant strains, confirming that Rsc2 is required for plasmid stability in our strain background (S4A Fig).

Chromosome localization of Rep1 and Rep2 was impacted in the *rsc2* mutant strains as seen by the reduced concordance of their genome-wide binding profiles (S4B Fig). However, the effect on Rep2 was less pronounced. In general, Rep1 binding to its usual chromosomal locales was severely impaired by *rsc2Δ*, as evidenced by the ChIP-seq and the CUT&Tag assays (Fig 4A and 4B). A reduction in binding, though smaller in magnitude, was seen for Rep2 as

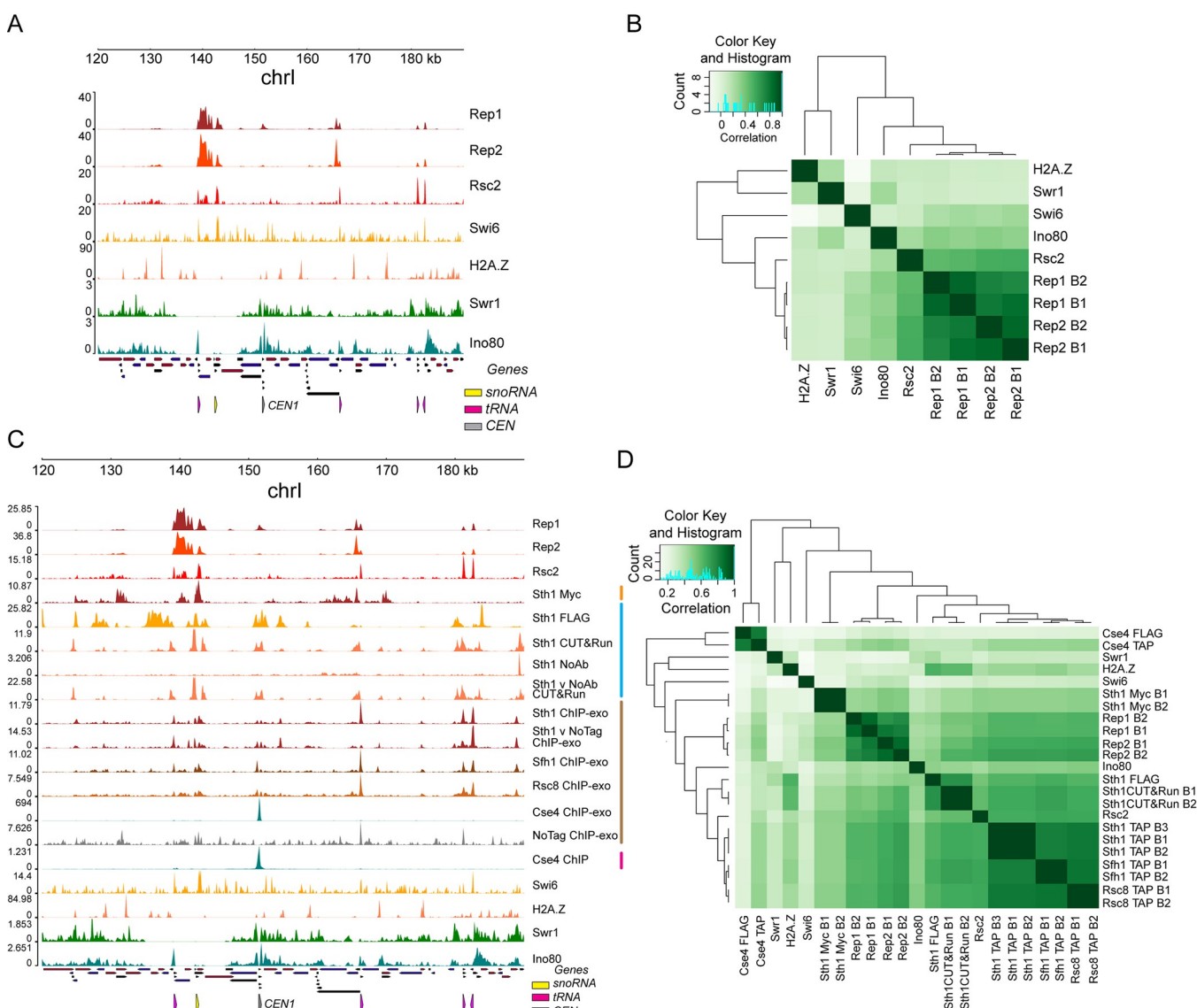

**Fig 3. Rep protein localization correlates with RSC2. (A)** Localization at a region of chrI of Rep1, Rep2, Rsc2 and unrelated chromatin factors, measured using ChIP-seq performed in our lab (present study) [38,44]. **(B)** Clustered correlation matrix showing genome-wide localization similarity between the indicated samples. B1 and B2 refer to biological replicates of the ChIP-seq experiments for Rep1 and Rep2. **(C)** Localization at a region of chrI of Rep1, Rep2, Rsc2 as well as other components of the RSC complex and unrelated chromatin factors performed either in our lab (present study) [38,44] or previously published studies from other labs, indicated by colored bars: orange [45], blue [46, 48], brown [47], magenta [49]. **(D)** Clustered correlation matrix showing genome-wide localization similarity between the indicated samples. B1, B2 and B3 refer to biological replicates.

well in both assays (Fig 4A and 4B). This differential effect of *rsc2Δ* on Rep1 versus Rep2 localization at genomic sites could be visualized using average binding profiles and heat maps (Fig 5A and 5D).

In contrast to the *rsc2Δ* effect, an increase in Rep1 chromosome association in the strain with the *rsc2* truncation was evident in the ChIP-seq profiles (Fig 4A), suggested by the CUT&Tag profiles (Fig 4B), and revealed in the binding enrichment-heat map representations (Fig 5A–5D). The disparate effects of *rsc2Δ* and *rsc2* truncation on Rep1 were not seen for Rep2 (Figs 4A, 4B and 5A–5D). However, there was evidence for a redistribution of Rep1-Rep2 around their normal localization sites (tRNA and snoRNA genes) (Figs 4A, 4B and S4C–S4E).

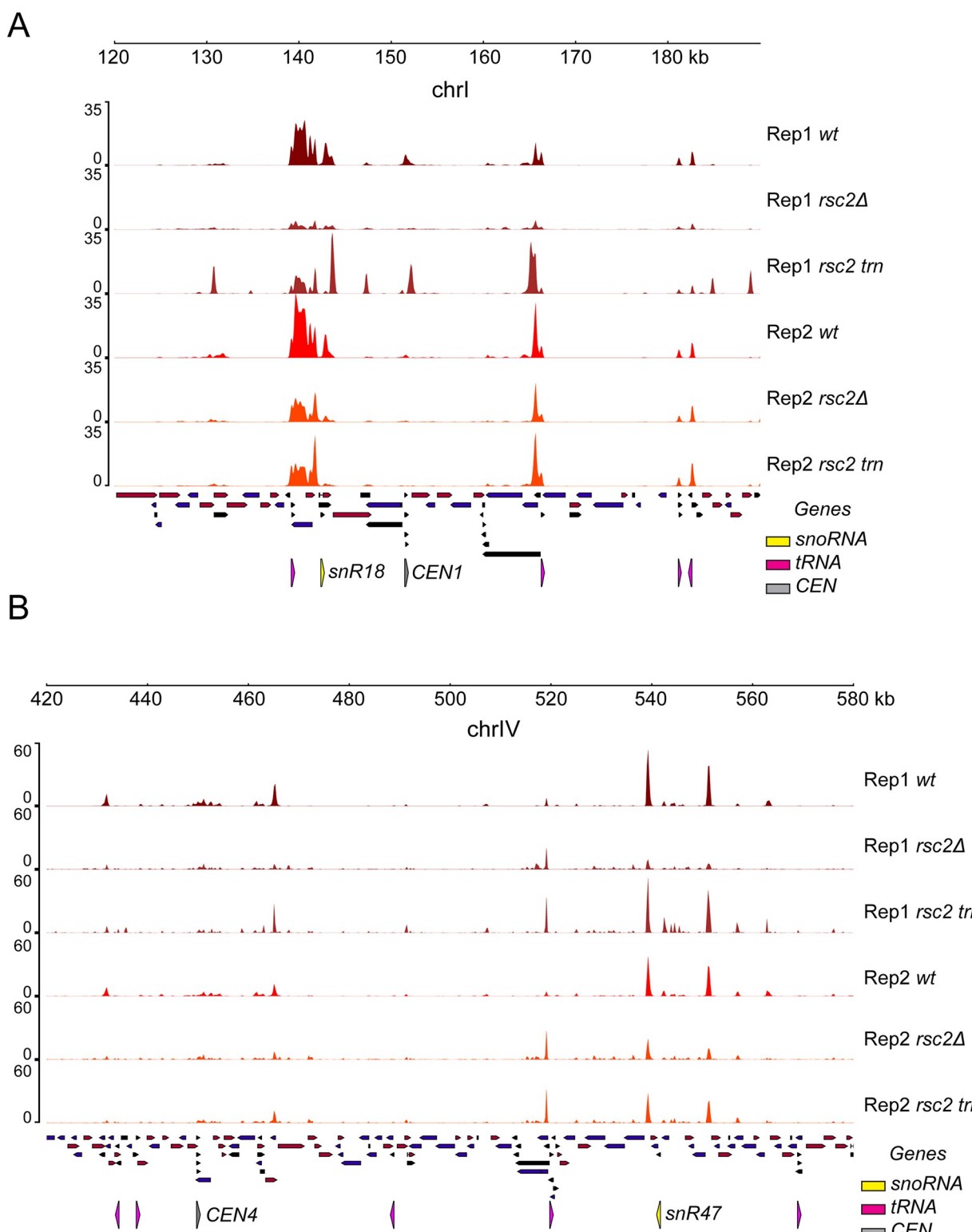

**Fig 4. Rep1 localization is affected by RSC.** (**A**) Rep1 and Rep2 localization at a region of chrI in wild type, *rsc2Δ* and truncated *rsc2* strains as measured by ChIP-seq. (**B**) Rep1 and Rep2 localization at a region of chrIV in wild type, *rsc2Δ* and truncated *rsc2* strains as measured by CUT&Tag.

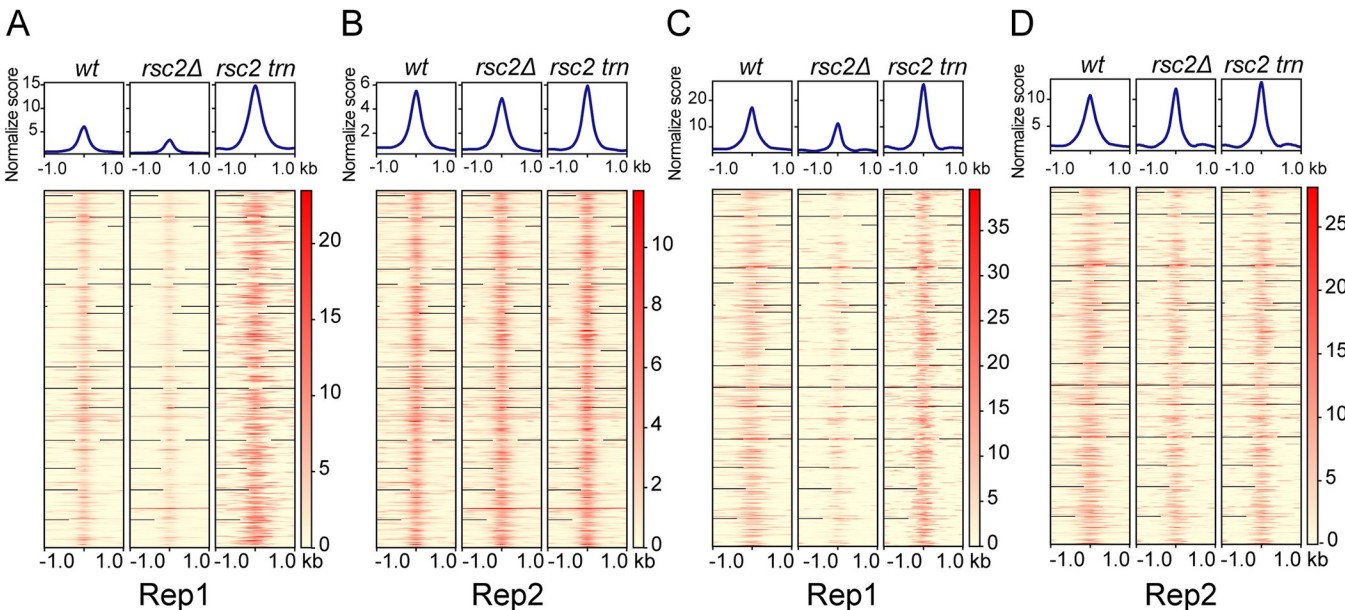

**Fig 5. Genomic localization of Rep1 is disrupted in *rsc2Δ*.** (A) Rep1 and (B) Rep2 localization as measured by ChIP-seq is shown in the strain backgrounds indicated across the top. In each profile plot or heatmap, the center is the midpoint of the Rep1 or Rep2 binding site identified in a wild type strain. (C) Rep1 and (D) Rep2 localization as measured by CUT&Tag.

This aberrant association at non-cognate sites could be responsible, at least in part, for the increased Rep1 binding due to *rsc2* truncation.

To examine the *rsc2Δ* and *rsc2* truncation effects in greater detail, we focused on the prominent types of Rep binding loci, namely tRNA loci, *TEL*s and *CEN*s. At tRNA loci, Rep1 binding as assayed by ChIP-seq was clearly diminished by *rsc2Δ* while it was disrupted to yield broader and less defined binding peaks by *rsc2* truncation (Fig 6A). In contrast, at *TEL*s, the strain with truncated *rsc2* showed more pronounced impairment of Rep1 localization than *rsc2Δ* (Fig 6B). At *CEN*s, the normally prominent peak of Rep1 localization was completely lost in both *rsc2Δ* and truncated *rsc2* strains, revealing the most profound impact of RSC2 on Rep1 localization (Fig 6C). At all three classes of sites, the loss of RSC2 showed little to no effect on Rep2 binding (S5A–S5C Fig).

We infer from these results that the manner in which Rep1 localizes to its chromosomal target sites varies among different classes of such sites, which can therefore be differentially affected by *rsc2Δ* or truncated *rsc2*. The reduced sensitivity of Rep2 to *rsc2* mutations suggest redundant mechanisms for chromosome localization of Rep1 and Rep2, whose effects on the individual proteins may not be equivalent.

## Rep1-Rep2 interactions with the RSC2 complex

Prior observations on *STB*-plasmid segregation, mother bias and plasmid association with chromosome spreads in the absence of functional Rsc2 [23,36] are consistent with plasmid-chromosome bridging through the interaction between *STB*-bound Rep1-Rep2 and the RSC2 complex. This notion is further supported by the baiting of RSC2 complex proteins by TAP-tagged Rep1 or Rep2 [35] and by the effects of *rsc2Δ* on the genome-wide localization of Rep1 (Figs 4–6). To better understand the role of the RSC2 complex in Rep1-Rep2 recruitment to chromosomes and 2-micron plasmid hitchhiking, we now characterized extensively Rep1-Rep2 interactions with the RSC2 complex by genetic tests, co-immunoprecipitation

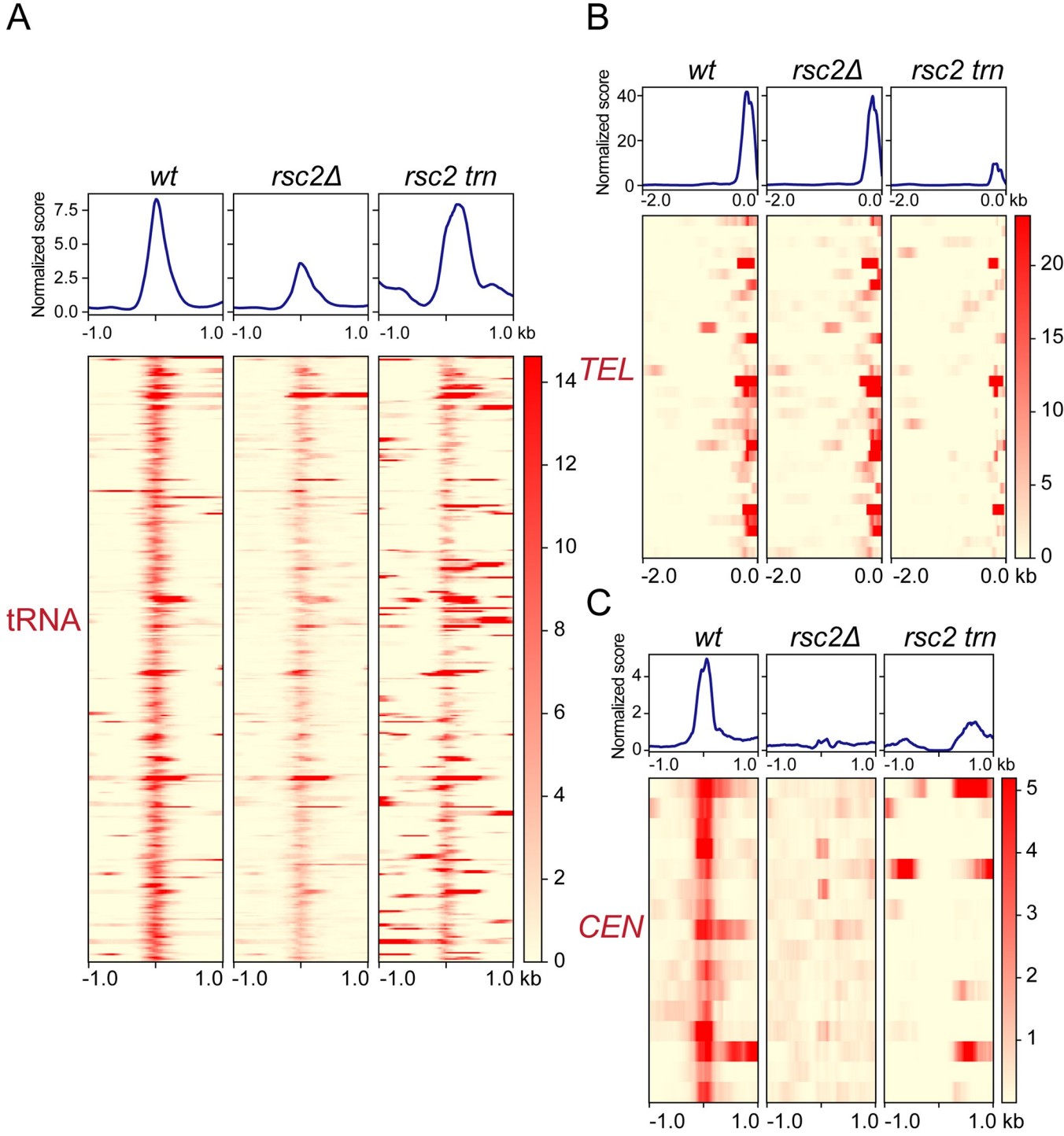

**Fig 6. Disruption of Rep1 localization at tRNA loci, *TEL*s and *CEN*s under *rsc2* mutation.** Binding of Rep1 at (**A**) tRNA loci, (**B**) *TEL*s and (**C**) *CEN*s as measured by ChIP-seq.

assays, and by western blot/mass-spectrometry analyses of the RSC1/RSC2 complex and of Rep1/Rep2 after affinity enrichment from yeast extracts.

A two-hybrid screen using a yeast cDNA library initially identified a truncated version of Sfh1, a common constituent of the RSC1 and RSC2 complexes, as an interacting partner of

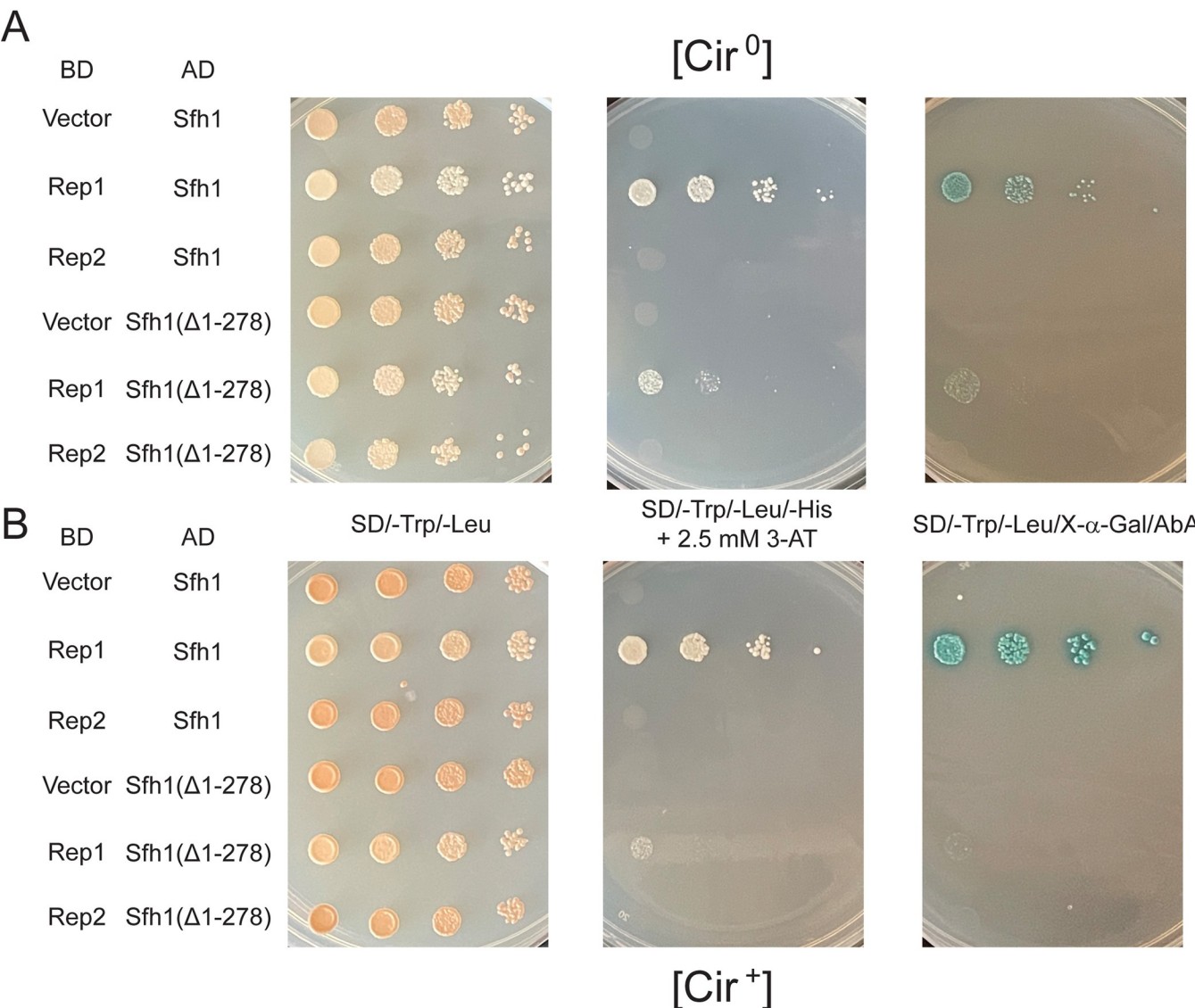

**Fig 7. *In vivo* interaction in yeast between the RSC complex component Sfh1 and Rep1 or Rep2. (A, B)** The listed protein pairs (or vector-protein controls) were tested according to the Gal4-based 'Matchmaker Two-Hybrid System' (Clontech) protocol. Positive interaction was assayed by the expression of three reporter genes, *HIS3* (in presence of 3-AT), *MEL1* (α-galactosidase; in presence of X-α-gal) and *AUR1-C* (resistance to Aureobasidin A). Serial dilutions (10x) of cultures grown under selection for the two plasmids expressing test partners were spotted on a plate without additional selection (left panel) or on the indicated reporter plates (middle and right panels). As Sfh1 itself is a modest transcriptional activator when bound to DNA, only one of the two 'bait'-"prey" combinations (with the activation domain fused to Sfh1) was tested. The [Cir+] and [Cir0] experimental strains were isogenic, except for the presence or absence of the native 2-micron plasmid, respectively. DB = DNA binding domain; AD = activation domain.

Rep1. Rep1 interaction with Sfh1 or its originally spotted deletion derivative Sfh1(Δ1–278) (aa-279 to aa-426) was observed in [Cir+] or [Cir0] host strains (Figs 7 and S6). The positive test in the [Cir0] background indicates that neither Rep2 nor the 2-micron plasmid (as a carrier of *STB*) is required for Rep1-Sfh1 interaction. By contrast, Rep2 interaction with Sfh1 or Sfh1 (Δ1–278) was not detected in the [Cir+] or the [Cir0] strain. These *in vivo* interactions in yeast were verified by reciprocal affinity pull-down of the proteins expressed in *E. coli* (Figs 8 and S7). This assay showed that the Sfh1 or Sfh1(Δ1–278) interaction was specific to Rep1 (Figs 8 and S7; lanes 4), with little to no interaction with Rep2 being detected. The full-length Sfh1

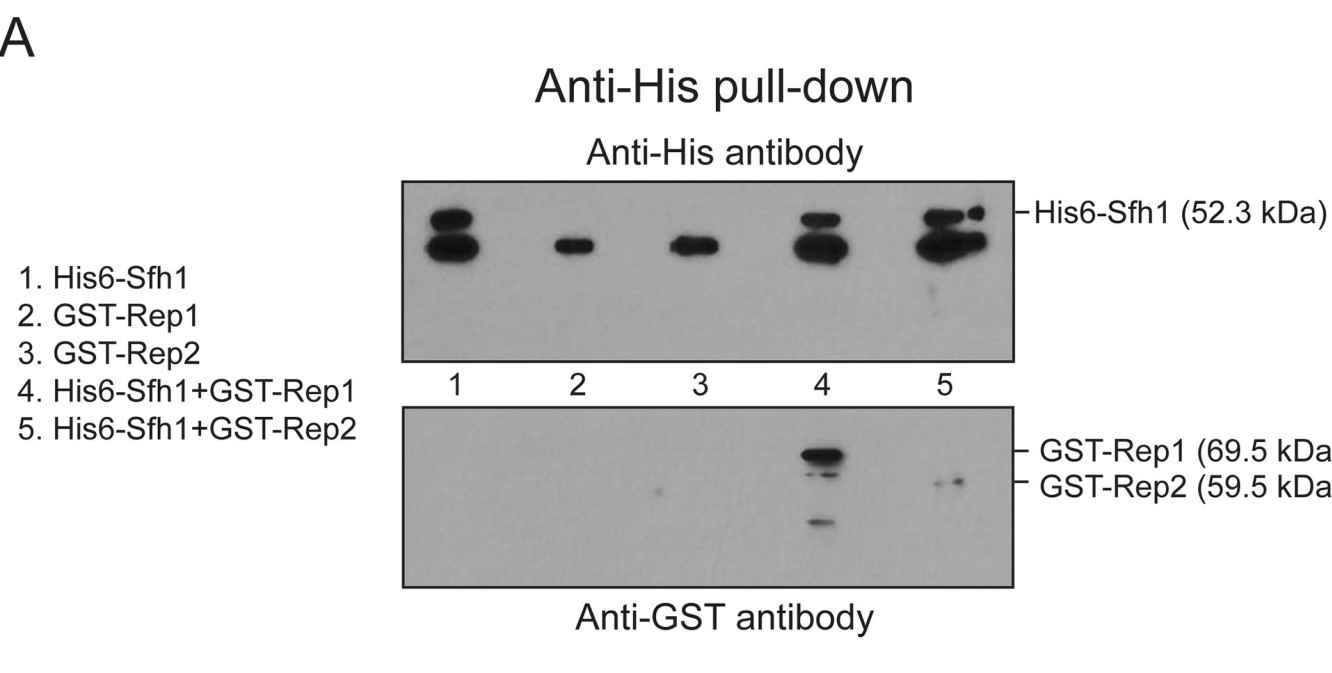

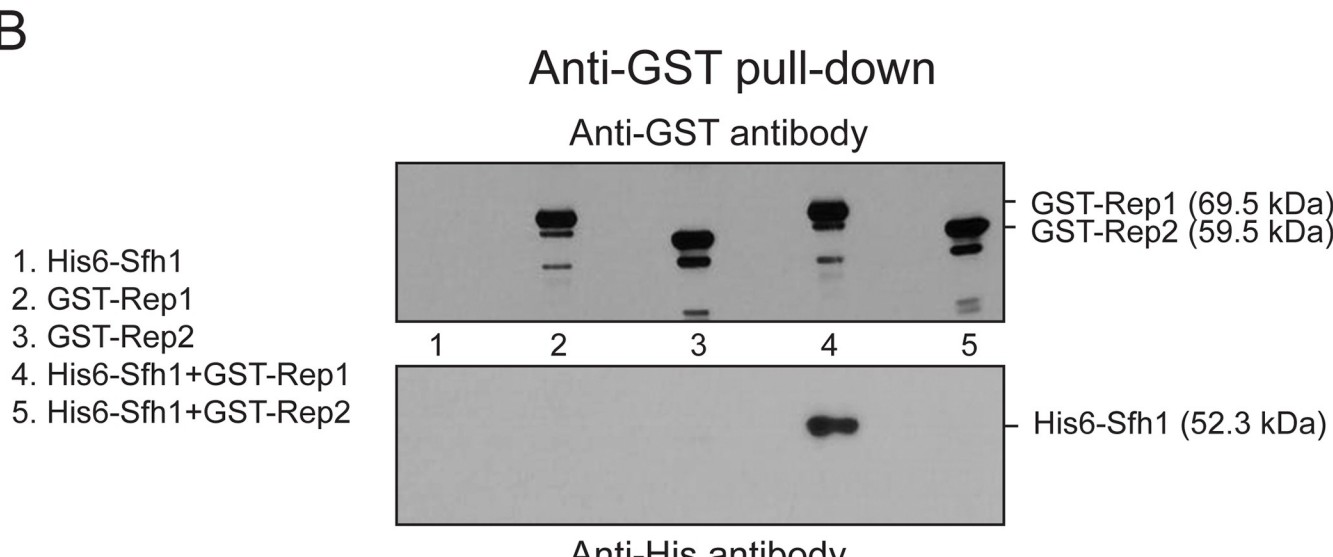

**Fig 8. Interaction of Sfh1 with Rep1 or Rep2 when these proteins are expressed in *E. coli*.** The indicated proteins were expressed in *E. coli* individually or in pairs, and were immunoprecipitated using Anti-His (**A**) or Anti-GST (**B**) antibodies. Co-immunoprecipitation of partner proteins was probed by western blot analysis.

showed marginal interaction with Rep2 (Fig 8; lanes 5) while Sfh1(Δ1–278) showed none (S7 Fig; lanes 5). However, co-expression of Rep1 with Rep2 resulted in more robust Rep2-Sfh1 interaction (S8 Fig; lanes 5 and 8). The association of Rep1 and Rep2 with affinity-enriched yeast RSC2 complex was shown by mass spectrometry (Fig 9; Table listing peptide-spectral count) and by western blotting (Fig 9; lanes 6 of the right panels). This tri-partite RSC2 complex-[Rep1-Rep2] interaction was validated by reciprocal affinity-enrichment of

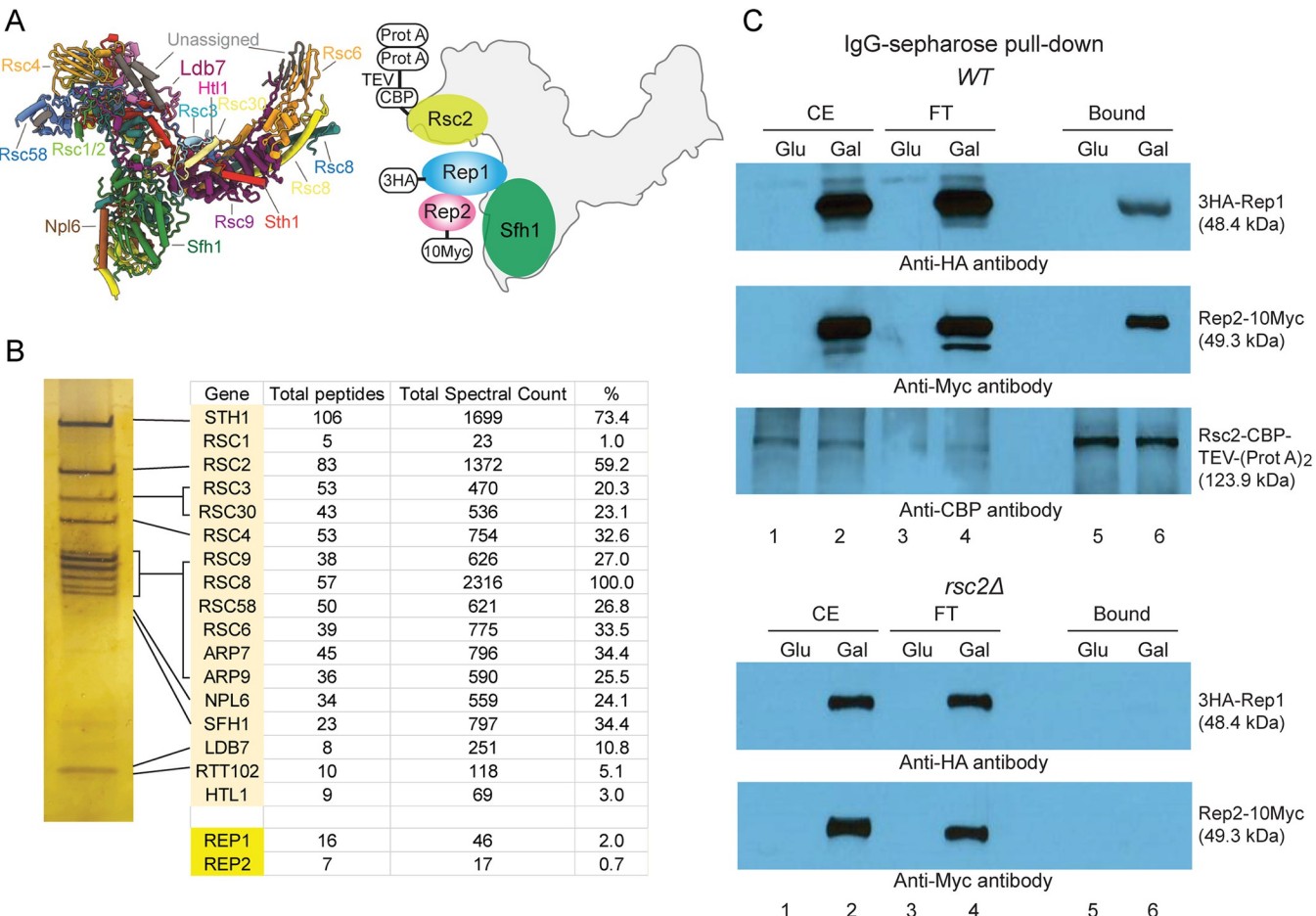

**Fig 9. Interactions of Rep1 or Rep2 with affinity-enriched RSC2 complex. (A)** The cryo-EM structure of the yeast RSC complex [86] (RCSB PDB 6V8O) and an idealized rendition of the potential interaction of Rep1-Rep2 with the RSC2 complex are diagrammed at the top left. The epitope tags fused to Rep1 (HA), Rep2 (Myc) and Rsc2 are indicated. The CBP (calmodulin binding peptide)-tag on Rsc2 was separated from the two tandem Prot A (Protein A)-tags by the TEV protease cleavage sequence. **(B)** A silver-stained profile of the RSC2 complex following affinity enrichment using IgG-sepharose and fractionation by SDS-polyacrylamide gel electrophoresis is shown on the left. The relevant mass spectrometry data are shown in the adjacent Table. **(C)** Results from western blot probing of the enriched RSC2 complex from a wild type (*WT*) strain using anti-HA and anti-Myc antibodies (to target Rep1 and Rep2, respectively). As a negative control, mock enrichment was performed in an *rsc2Δ* strain. In the assays shown here and in S9–S11 Figs, the expression of the tagged Rep1 and Rep2 was controlled by the *GAL* promoter.

Rep1 and Rep2 as well or by enrichment of the RSC2 complex using an alternative affinity trap (S9 Fig; lanes 6).

Taken together, the interaction results are consistent with independent interaction of Rep1 and Rep1-dependent interaction of Rep2 with the RSC2 complex. They also fall in line with the more marked disruption in the chromosome localization of Rep1 than Rep2 by *rsc2Δ* or *rsc2* truncation, as seen in ChIP-seq and CUT&Tag mapping (Figs 4–6). However, we cannot rule out Rep2 interaction with the RSC2 complex via a subunit protein other than Sfh1 or exclude Rep1 interaction with more than one component of the RSC2 complex. The lack of Rep2-Sfh1 interaction by the two-hybrid test in the [Cir⁺] strain (expressing Rep1 from the 2-micron plasmid) is surprising, as Rep1 and Rep2 are strong interacting partners [41,42,51] (S6 Fig). It is possible that Rep1-Sfh1 interaction or the DNA binding domain fused to Rep2 masks (at least partially) the protein interface required for Rep1-Rep2 interaction (see S6 Fig and its legend). Alternatively, transcriptional activation from the Rep1-bridged interaction

between Rep2 and Sfh1 might fall short of the selection thresholds employed in the two-hybrid assays.

The interaction data, together with the ChIP-seq and CUT&Tag results, validate the role of [Rep1-Rep2]-RSC2 association in chromosome tethering and chromosome-like segregation of the 2-micron plasmid. Association of Rep1-Rep2 with the enriched RSC1 complex as well (S10 and S11 Figs) is not surprising, as Sfh1 is one of the fifteen proteins common to both complexes. However, this incidental interaction does not contribute to plasmid stability [36].

## Localization of Rep1, Rep2 and an *STB*-plasmid with respect to Rsc1 and Rsc2 in yeast chromosome spreads

The prevalence of common chromosome localization sites for Rep1-Rep2 and the RSC2 complex revealed by genomics (Fig 3), the interaction of Rep1-Rep2 with the RSC2 complex (Figs 7–9 and S6–S11) [35], and the established role of the RSC2 complex in 2-micron plasmid stability [35,36] (S4A Fig) imply the presence of Rep1-Rep2 and an associated *STB*-plasmid at a subset of the chromosome locales occupied by this chromatin remodeler in individual yeast nuclei. Conversely, given the dispensability of Rsc1 for plasmid segregation [36], no colocalization of the Rep proteins or an *STB*-plasmid with the RSC1 complex on chromosomes is anticipated. We tested these predictions by assaying the localization of Rep1, Rep2 and an *STB*-plasmid with respect to Rsc1 or Rsc2 in mitotic and meiotic chromosome spreads.

In mitotic spreads from a [Cir$^+$] strain, there was a high correlation in the chromosome colocalization of Rep1 or Rep2 with Rsc2 (Figs 10A, 10B and S12A). A multi-copy *STB*-reporter plasmid also showed a similar correlation (Figs 10C, 10D and S12A), as expected from the tight association of the partitioning proteins with single copy *STB*-plasmids or with clusters of multi-copy *STB*-plasmids in yeast nuclei or in chromosome spreads [23,43]. Rsc1, by contrast to Rsc2, showed no strong colocalization with Rep1, Rep2 or the *STB*-plasmid in mitotic spreads (Figs 10A–10D and S12B). The significant overlap between Rsc1 and Rsc2 seen in mitotic chromosome spreads (S13 Fig) was not surprising, as the RSC1 and RSC2 complexes possess unique as well as shared functions [34]. Despite this substantial Rsc1-Rsc2 coalescence, the strong Rep1-Rep2 bias towards Rsc2 further reinforces the specific role of the RSC2 complex in 2-micron plasmid partitioning.

In [Cir$^0$] strains expressing only one of the Rep proteins, there was a sharp reduction in the fraction of mitotic chromosome spreads containing Rep1 or Rep2 foci as well as in the number of Rep foci per spread. Furthermore, these few foci showed poor correlation in their association with Rsc1 or Rsc2 (S14 Fig). Meiotic chromosome spreads verified the preferential colocalization of Rep1 or Rep2 with Rsc2 observed in the [Cir$^+$] spreads, and further demonstrated the restoration of Rsc2-Rep colocalization in [Cir$^0$] spreads by Rep1-Rep2 mutual complementation (S15 Fig). These results agree with earlier findings that efficient chromosome association of Rep1 and Rep2 and that of an *STB*-plasmid require both partitioning proteins [52].

Overall, the localization data corroborate the inference from the genomics and interaction assays that the RSC2 complex is a key mediator, even if it is not the only one, in tethering the 2-micron plasmid to yeast chromosomes. Potential redundancy in host factors that promote chromosome association would be beneficial to the 2-micron plasmid or to any selfish DNA element whose stability is dependent on chromosome-hitchhiking.

## Nuclear localization of the 2-micron plasmid near tRNA genes

In an earlier study using a nearly single-copy *STB*-plasmid (pSG1-*STB*), 50–60% of the plasmid positions within the nucleus could be accounted for by *TEL*- and *CEN*-proximity [37]. The presence of additional chromosome sites that license plasmid tethering, as suggested by this

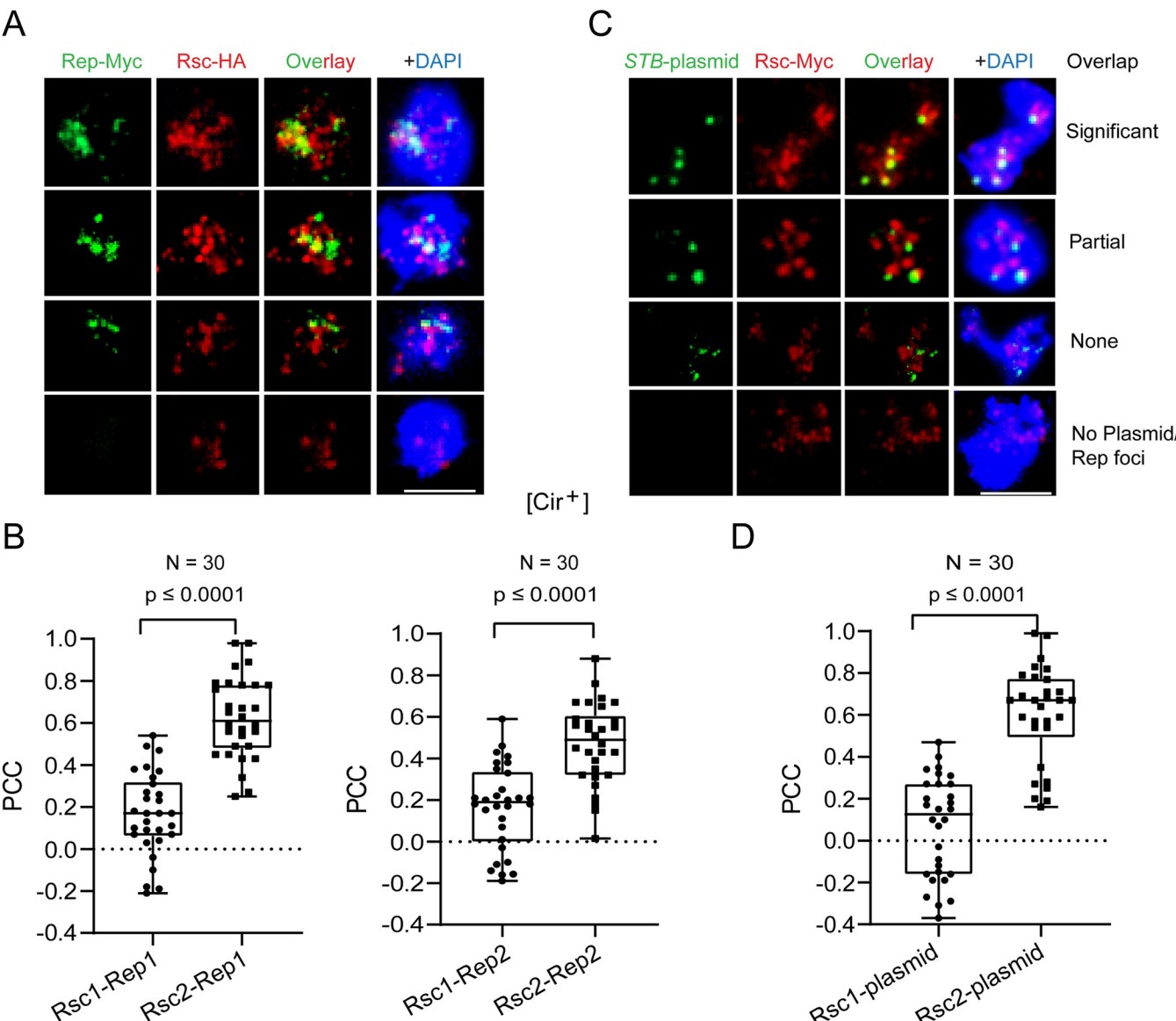

**Fig 10. Localization of Rep1, Rep2, and an *STB*-reporter plasmid with respect to Rsc1 or Rsc2 in mitotic chromosome spreads.** Chromosome spreads were prepared from diploid [Cir$^+$] mitotic cells. **(A, B)** HA-tagged Rsc1 or Rsc2 was expressed from the modified chromosomal *RSC1* or *RSC2* locus, respectively, along with Myc-tagged Rep1 or Rep2 from an integrated cassette. Rep expression was under *GAL* promoter control. **(C, D)** The *STB*-plasmid containing a [LacO]$_{256}$ array was present in a strain expressing GFP-LacI from the *HIS3* promoter and Myc-tagged Rsc1 or Rsc2 from the native *RSC1/RSC2* promoter. **(A-D)** Rep1, Rep2, Rsc1 and Rsc2 were visualized by immunofluorescence using antibodies to the respective epitope tags attached to them. The plasmid was visualized indirectly by the bound GFP-LacI using an antibody to GFP. Typical images of the four categories of spreads analyzed are shown at the top **(A, C)** The degree of overlap between the red and green fluorescence was estimated as the Pearson's correlation coefficient (PCC) and graphed as box plots at the bottom **(B, D)** The 'Significant', 'Partial' and 'None' or 'No Plasmid/Rep foci' types correspond to PCC ranges of 0.5 to 1.0, > 0.3 to < 0.5, and ≤ 0.3. Bar = 5 μm.

analysis, has been borne out by the present genome-wide ChIP-seq and CUT&Tag assays (Figs 1 and 4). A particular advantage of the pSG1 reporter plasmid is that it could be manipulated to behave as a *CEN*-plasmid (pSG1-*CEN*), an *STB*-plasmid (pSG1-*STB*) or as an *ARS*-plasmid (pSG1-*ARS*) by appropriately altering the experimental conditions [33] (S16 Fig). The three versions of pSG1 are thus well suited for comparing plasmid localization and segregation promoted by *CEN* versus *STB* partitioning systems against those in the absence of either (*ARS*). In

mitotic yeast cells, *CEN*s are congressed into a single tight-knit cluster, *TEL*s are assembled as three to six less cohesive clusters, and tRNA coding loci (~300 per haploid genome) are distributed in two clusters, one of which localizes with *CEN*s and the other with the rDNA array at the nucleolus [53–57]. Prompted by the current ChIP-seq and CUT&Tag results showing Rep1-Rep2 proclivity to localize at tRNA genes (Figs 1 and 2), and given the potential spatial ambiguity in the nuclear positions of *CEN* and tRNA loci, we probed the localization of pSG1-*CEN*, pSG1-*STB* and pSG1-*ARS* with respect to tRNA$_{val-UAC}$ locus on chrIV (*CEN4*, 449711; *tV(UAC)D*, 488797). Analysis of published Hi-C chromosome conformation capture data reveal that the tRNA$_{val-UAC}$ locus, which is about 39 kb from *CEN4*, is part of the *CEN*-proximal tRNA gene cluster and not the nucleolar tRNA gene cluster [57], consistent with a block in pol III-transcribed genes near *CEN*s from associating with the nucleolus [58]. Tighter association of pSG1-*CEN* than pSG1-*STB* with the tRNA$_{val-UAC}$ locus is expected, as tRNA loci are only one set of preferred tethering sites for pSG1-*STB*. By contrast, any association of pSG1-*ARS*, which does not localize to chromosomes, with tRNA genes can only be a chance occurrence.

The pSG1-*CEN* plasmid foci were in closer proximity to the tRNA$_{val-UAC}$ locus than pSG1-*STB* foci, as indicated by the median distance plots (Fig 11). The cluster of tRNA loci containing the tRNA$_{val-UAC}$ gene would be associated with the single *CEN*-cluster, which includes the pSG1-borne *CEN* as well. By comparison, the pSG1-*STB* foci showed a less compact distribution, yet with a subset of the plasmid foci still positioned near the tRNA$_{val-UAC}$ locus ($\leq$ 0.5 μm; Fig 11). The distribution of pSG1-*ARS*, which is known to be largely absent from chromosome spreads [23], was more or less random across the nucleus (Fig 11). The median value for the pSG1-*ARS* to tRNA$_{val-UAC}$ gene spacings was centered around 1 μm, which is expected if the plasmid has no preferred location within the ~2 μm diameter nucleus and the reference tRNA locus is at the nuclear periphery by association with the *CEN* cluster.

The relative proximity of pSG1-*CEN* versus pSG1-*STB* to the tRNA$_{val-UAC}$ locus is consistent with the collective results from cell biology, ChIP-seq and CUT&Tag-seq. The 2-micron plasmid is often present at or near *CEN*s, *TEL*s and tRNA loci (as suggested by all three assays combined); however, plasmid-chromosome association is not limited to these loci alone (as deduced from the genome-wide analyses).

## A Rep1-Rep2 bypass system for plasmid association with the RSC2 complex promotes segregation by chromosome-hitchhiking

The cumulative findings from the present study, complemented by relevant prior observations, suggest RSC-promoted plasmid association with chromosomes to be a central step in plasmid segregation by chromosome-hitchhiking. In principle then, an *ARS*-plasmid non-covalently linked to the RSC complex should be able to overcome mother bias and improve its equal segregation frequency. In order to test this possibility, we relied on a [repressor-operator]-based experimental design described previously for recruiting proteins to specific sequences on plasmids to follow their segregation [59–61].

We introduced the nearly single-copy pSG1-*ARS* (which harbors a [LacO]$_{256}$ array; S16 Fig) into a [Cir$^0$] yeast strain engineered to express GFP-LacI and Sfh1-LacI simultaneously. We reasoned that that both GFP-LacI and Sfh1-LacI would be present on the plasmid as the two fusion proteins are expected to bind the operators with roughly equal affinity. The bound Sfh1-LacI would promote non-covalent association of pSG1-*ARS* with the RSC complex, while the bound GFP-LacI would provide the fluorescence tag for plasmid visualization. The equal segregation of pSG1-*ARS* in a single generation assay showed a significant increase in the strain expressing Sfh1-LacI compared to the control strain expressing native Sfh1 (Fig 12A). In

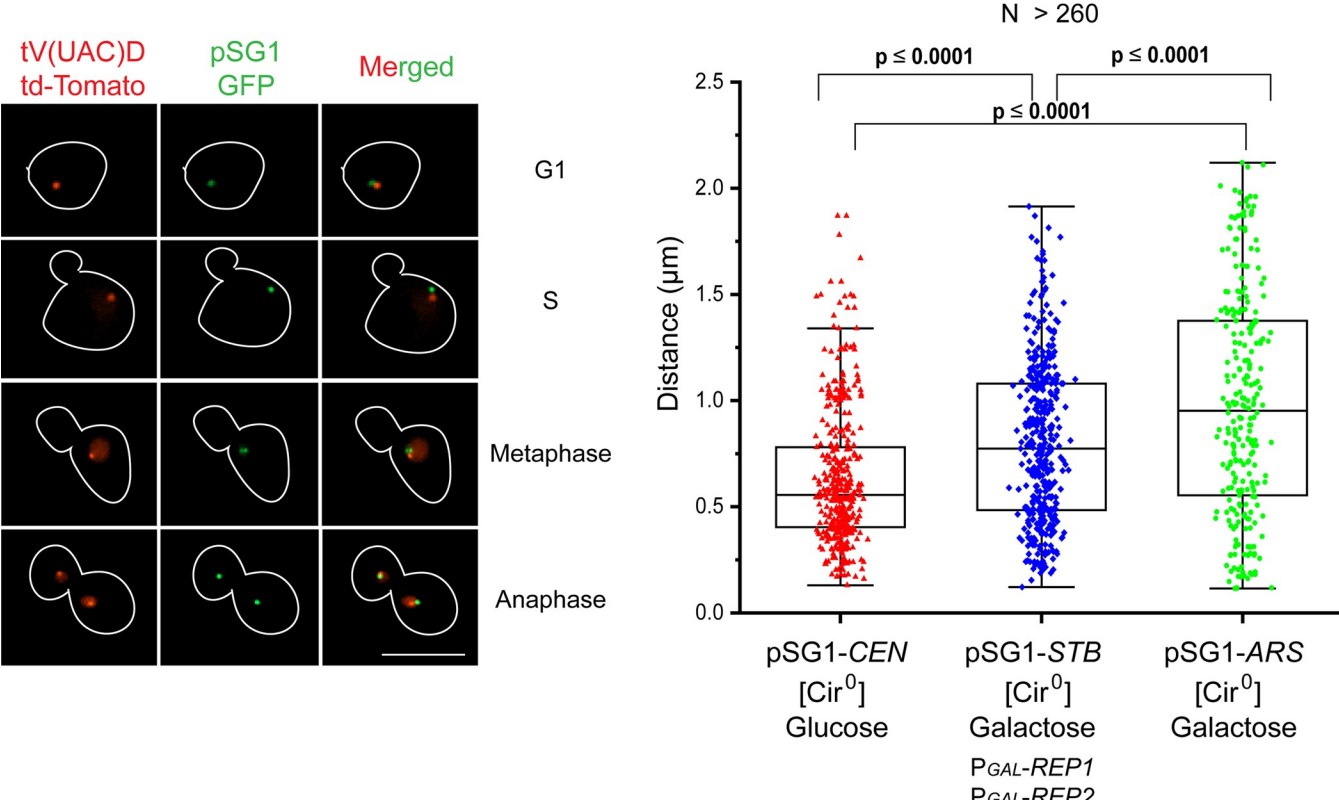

**Fig 11. Nuclear localization of pSG1-*CEN*, pSG1-*STB* or pSG1-*ARS* with reference to a tRNA gene.** For visualizing the plasmid and tRNA loci simultaneously, pSG1-([LacO]$_{256}$) ([33]; S16 Fig) was introduced into two closely matched [Cir$^0$] strains in which the tRNA$_{val-UAC}$ locus (chromosome IV; 488797) was tagged by an adjacent [TetO]$_{\sim50}$ array and GFP-LacI and TetR-[td-Tomato] were constitutively expressed. One of these strains contained an integrated cassette for the inducible expression of Rep1 and Rep2 from the *GAL* promoter. Only G1 cells containing a single green plasmid focus (~90%) and cycling cells containing one red focus of the tagged tRNA gene and no more than two green foci per nucleus were included in the analysis. Depending on the functional state of *CEN* or *STB* in pSG1, replicated plasmid sisters in a nucleus could be coalesced into a single focus or form a pair of overlapping or resolved foci. The paired tRNA sister loci would appear as one focus until their segregation during anaphase. Typical images of plasmid and tRNA gene foci in cells at different cell cycle stages are shown at the left. Not included here is the subset of anaphase cells with one of the nuclei lacking green fluorescence due to plasmid missegregation. Distances measured between the centroids of the green and red foci within a cell (or a cell compartment) are plotted at the right. When two plasmid foci were present in a nucleus, their average distance from the tRNA gene focus was taken. The data correspond to over 260 cells scored for each plasmid type. pSG1-*CEN*: [Cir$^0$], P$_{GAL}$-*REP1*, P$_{GAL}$-*REP2*, glucose; pSG1-*STB*: [Cir$^0$], P$_{GAL}$-*REP1*, P$_{GAL}$-*REP2*, galactose; pSG1-*ARS*: [Cir$^0$], galactose. Bar = 5 μm.

addition, Sfh1-LacI partially relieved the mother bias associated with plasmid missegregation (Fig 12A). Furthermore, the negligible association of this plasmid with chromosome spreads from the host strain expressing Sfh1 was significantly boosted when Sfh1-LacI replaced Sfh1 (Fig 12B). The results with pSG1-*ARS* were confirmed by applying the same [operator-repressor]-mediated Sfh1 recruitment strategy to a multi-copy *STB*-[LacO]$_{256}$-plasmid harbored by a [Cir$^0$] strain in a longer-term stability assay (Fig 12C). Consistent with increased plasmid stability, Sfh1-LacI also substantially increased plasmid localization in chromosome spreads compared to Sfh1 (Fig 12D).

Thus, artificial plasmid-chromosome association with the RSC complex as the tethering agent can recapitulate plasmid segregation by hitchhiking, albeit at a lower efficiency than that of the native Rep-*STB* based hitchhiking system.

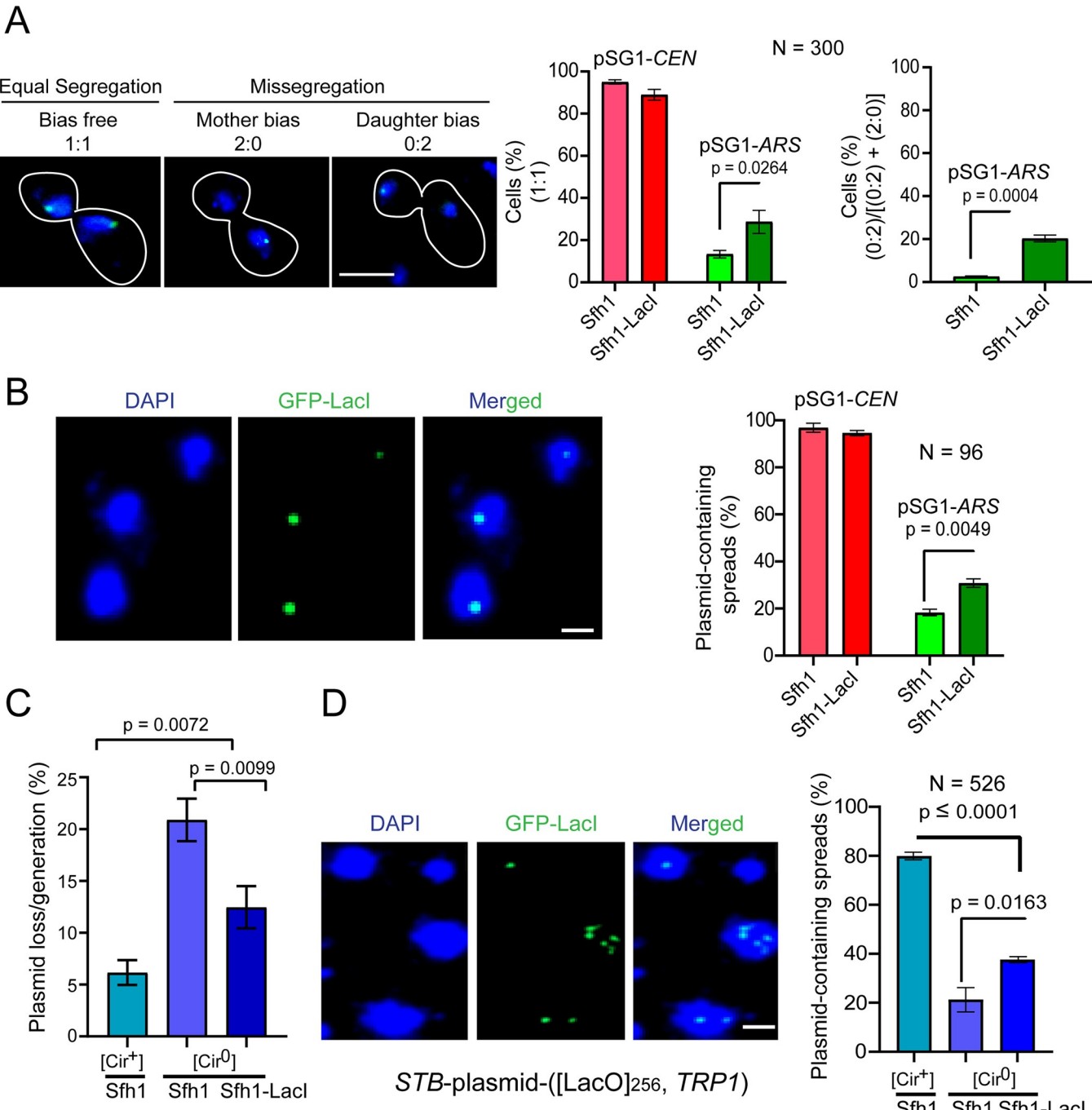

**Fig 12. Sfh1-LacI improves equal segregation, partially relieves mother bias, and augments chromosome association of *ARS*-plasmids containing [LacO] arrays.** (A) Plasmid pSG1-([LacO]$_{256}$) was manipulated to behave as pSG1-*CEN* (glucose) or pSG1-*ARS* (galactose) in haploid [Cir$^0$] strains expressing GFP-LacI (from the *HIS3* promoter) or GFP-LacI along with Sfh1-LacI, which was controlled by the *SFH1* chromosomal promoter. Cells arrested in G1 with α-factor were released into the cell cycle, and plasmid segregation was followed in late anaphase cells. Frequencies of equal segregation (1:1) and missegregation into the daughter cell (0:2), as a fraction of all missegregation events (2:0 plus 0:2), are plotted. (B) In chromosome spreads prepared from G1-arrested cells of the two strains indicated under (A), plasmid foci were visualized by immunofluorescence using a GFP-directed antibody. (C) Plasmid loss rate under non-selective growth was assayed in the same strains as in (A) and (B) harboring a multi-copy *STB*-plasmid-([LacO]$_{256}$, *TRP1*) in place of pSG1. This reporter plasmid is functionally an *ARS*-plasmid in a [Cir$^0$] host, as Rep1 and Rep2 are absent. As a control, plasmid stability was measured in a [Cir$^+$] strain expressing GFP-LacI and Sfh1. (D) Chromosome spreads from G1-arrested cells described under C were assayed for the presence of the plasmid. (A) Bar = 5 μm. (B, D) Bar = 2 μm.

## Discussion

Our findings shed new light on the mechanisms by which the yeast 2-micron plasmid, an exemplary selfish DNA element, hitchhikes on the chromosomes of its host for equal segregation. Although RSC2 was previously identified as a host factor for 2-micron plasmid stability, the mechanism of its action remained unknown. We show that RSC2 is responsible for plasmid-chromosome bridging by interacting with *STB*-bound Rep1 and Rep2. The plasmid's enlistment to chromosomes by the association of its partitioning proteins with a chromatin remodeling complex signifies a new mode of chromosome-hitchhiking among selfish DNA elements.

### Rep1-Rep2 and the RSC2 chromatin remodeling complex share chromosome locales by protein-protein interaction

The Rep1-Rep2 partitioning proteins co-occupy hundreds of sites on chromosomes but display a preference for tRNA, snoRNA, *CEN* and *TEL* loci. In a given nucleus, a small subset of these sites will serve as tethering sites for the three to six clusters of the multi-copy plasmid whose *STB* partitioning locus is bound by Rep1-Rep2. We provide multiple lines of evidence to demonstrate that the host factor (or at least a prominent one) responsible for plasmid-chromosome bridging is the yeast chromatin remodeling complex RSC2. First, there is strong overlap between the chromosome localization profiles of Rep1-Rep2 and the RSC2 complex, as revealed by genomics and cell biology. Second, the normal chromosome localization pattern of Rep1 is diminished as well as disrupted when the RSC2 complex is non-functional. The remnant non-functional or quasi-functional deposition of the partitioning proteins on chromosomes may result from their interaction with RSC component(s) other than Rsc2 or with partially assembled form(s) of the RSC complex. Alternatively, or in addition, differences in the global chromatin states among the wild type, *rsc2Δ* and *rsc2* truncated strains could, at least in part, account for the rather complex localization patterns of Rep1 seen in the mutant strains. Third, the Rep1-Rep2 proteins associate with the RSC2 complex by multiple criteria, including two-hybrid tests, immunoprecipitation assays, and affinity enrichment-mass spectrometry analyses. Thus, [Rep1-Rep2]-RSC2 interaction readily accounts for the concurrence between the chromosome occupancy of the RSC2 complex and Rep1-Rep2, and indirectly the 2-micron plasmid.

### Physical interaction of Rep1-Rep2 does not discriminate between the RSC1 and RSC2 complexes, yet only RSC2 interaction is functional

We also detect Rep1-Rep2 interaction with the RSC1 complex, which is identical to the RSC2 complex except in a single component (Rsc1 versus Rsc2). The two are redundant in many but not all their physiological roles [34]. Normal plasmid stability in the absence of Rsc1 reported previously [36] and distinct chromosome localization of Rsc1 and Rep1, Rep2 or an *STB*-plasmid seen in this study suggest that Rep1-Rep2 interaction with Rsc1 is not functionally relevant to 2-micron plasmid segregation [36]. Rep1-Rep2 interaction with the RSC1 complex may be limited to its chromatin-free pool or the Rep proteins may be dislodged from the complex upon its association with chromosomes. There is precedence for the selective utilization of one of two redundant host factors by the plasmid for its propagation. Of the nuclear kinesin motors Cin8 or Kip1, either of which is dispensable with no apparent ill effects on spindle integrity and chromosome segregation, only Kip1 is required for plasmid segregation [62]. The dependence by the plasmid on Rsc2 but not Rsc1 and Kip1 but not Cin8 is consistent with a selfish element moderating its selfishness in order to avoid fitness conflicts with its host.

Equal segregation of an *STB*-plasmid in the absence of Rsc2 is still considerably higher than that of an equivalent *ARS*-plasmid, and the mother bias is lower [23]. It is likely that the Rep-*STB* system enlists more than one host factor for plasmid-chromosome association. Alternatively, residual components of the RSC2 complex may be sufficient for inefficient or aberrant localization of Rep1-Rep2 on chromosomes, as suggested by ChIP-seq and CUT&Tag (Figs 4–6). Random plasmid tethering to chromosomes in the absence of Rsc2 can still counteract mother bias. However, equal segregation will be limited to 50% in agreement with the random assortment of chromosomes. The observed 1:1 segregation frequencies of a single-copy *STB*-plasmid in the wild type and *rsc2Δ* backgrounds (~67% versus ~47%, respectively) is consistent with this model [23].

## Clustered chromosome loci are among favored plasmid tethering sites

Interestingly, chromosome loci preferred by Rep1-Rep2, despite their physical separation or lack of linkage, tend to organize into clusters in the 3D nucleus of mitotic yeast cells, as does the multi-copy 2-micron plasmid as well. There is a single *CEN* cluster, three to six *TEL* clusters and two tRNA gene clusters, one associated with *CEN* and the other with the nucleolus [53–57]. The plasmid forms three to six clusters in a haploid nucleus, which are often present in proximity to the clustered chromosome loci. The RSC complex is required for *CEN*-clustering [63], and likely affects telomere clustering indirectly by its role in maintaining nuclear envelope structure and telomere anchoring [64]. The tRNA loci, widely dispersed in the genome and notable for the preferential localization of the RSC2 complex, influence long-range chromosome interactions, including *CEN*-clustering, presumably by their role in local chromatin architecture [65,66]. Our genome-wide data suggest a model in which clustered chromosome loci are preferred hitchhiking sites for the 2-micron plasmid as a result of their occupancy by Rep1-Rep2 with assistance from the RSC2 complex.

## A possible role for the RSC2 complex in promoting sister-to-sister plasmid segregation

A striking feature of 2-micron segregation, revealed by single-copy *STB*-plasmids, is the tethering of plasmid sisters formed by replication to sister chromatids to mimic chromosome-like one-to-one segregation [23,24]. The structurally related cohesin and condensin complexes, which act collaboratively to orchestrate chromosome compaction, sister chromatid pairing and their dynamics during segregation [67–69], associate with *STB* and assist 2-micron plasmid segregation as well [37,52,70]. The RSC complex facilitates condensin and cohesin loading on chromosomes, while also regulating differential association of cohesin with centromeres and chromosome arms [71–73]. Pairing of plasmid sisters and their tethering to sister chromatids may also require the combined action of all three complexes. It is possible that this crucial step in equal plasmid segregation is blocked even when plasmid chromosome association is not completely eliminated by *rsc2Δ*.

Can tethering of an *ARS*-plasmid to a chromatin remodeling complex other than RSC2, to histones or to any chromosome associated protein, support segregation by chromosome hitchhiking? The answer is likely yes, although the efficiency of plasmid segregation may vary depending on which chromatin binding protein or protein complex localizes the plasmid on chromosomes. Any chromosome associated moiety is expected to segregate to daughter cells in accordance with the random assortment of chromosomes. However, such segregation would be quite different from the one-to-one segregation of 2-micron plasmid sisters by hitchhiking on sister chromatids. Multiple host factors, including the RSC2 and cohesin complexes, are likely responsible for orchestrating this special type of chromosome hitchhiking. We have

not tested whether equal segregation of an *ARS*-plasmid is improved by tethering to non-RSC2 chromosome binding proteins as we were concerned that such an attempt is likely to fall within the class of experiments that is 'doomed to succeed'.

## A more general role for chromatin remodeling complexes in promoting chromosome coupled segregation of selfish DNA elements

Analogous to the 2-micron plasmid, episomes of mammalian gammaherpes and papilloma viruses also make use of chromosome hitchhiking for their stable maintenance in latently infected cells [25,26,28,30–32,74]. This common strategy utilized by disparate selfish DNA elements residing in hosts that diverged ~1.5 billion years ago presumably reflects convergent evolution of survival mechanisms that take advantage of the high fidelity of chromosome segregation. In fact, the origins of chromosome dependent segregation may be traced much farther back in evolution to prokaryotic plasmids that passively attach to the bacterial chromosome or actively surf on the nucleoid [75,76]. Whereas chromosome hitchhiking enables the 2-micron plasmid to overcome the diffusion barrier posed by the yeast nuclear architecture, it prevents the mammalian viral episomes from being exiled in the cytoplasm during the disassembly and reassembly of the nuclear membrane associated with cell division. The plasmid and viral systems appear to have arrived at the common solution of chromosome association in response to the contrasting challenges of closed mitosis in fungi and open mitosis in higher eukaryotes. Chromosome tethering of viral episomes may be effected via direct DNA interaction by a viral partitioning protein or through indirect interaction mediated by chromatin associated or chromatin component proteins that include hEBP, Brd4 and the histones H2A and H2B [25,31,32,77,78]. The exploitation of a chromosome remodeling factor by a selfish DNA element for chromosome attachment has no precedent. However, the presence of SWI/SNF class of chromatin remodelers analogous to the RSC complex in higher eukaryotes [79,80] raises the possibility that they may also be targeted by viruses or other selfish DNA elements for hitching a ride on chromosomes.

## Materials and methods

### Yeast strains and plasmids

The yeast strains used in this study and their relevant genotypes are listed in S1 Table. Strains containing or lacking the native 2-micron plasmid are referred to as [Cir$^+$] or [Cir$^0$], respectively. The [Cir$^+$] designation does not include [Cir$^0$] strains transformed with 2-micron derived or other artificial plasmid constructs. The plasmids employed in this work and their salient features are summarized in S2 Table.

### Integration of expression cassettes into chromosomes, other genome modifications

Yeast chromosome loci were engineered as described previously by integration of linearized plasmids cut within regions of homology, insertion of PCR amplified DNA fragments with flanking homology to desired chromosome sequences or by using CRISPR (Cas9-sgRNA) technology [12].

### Chromatin immunoprecipitation and sequencing

Chromatin immunoprecipitation was done as previously described [81]. Briefly, 300 ml of log-phase yeast culture was induced with 2% galactose for 2 hours at 30˚C. Chromatin was crosslinked by addition of formaldehyde to a final concentration of 1% and incubation at 25˚C for

30 min, shaking at 100 rpm. Crosslinking was quenched by addition of glycine (final 125 mM) and shaking for 5 min. Cells were harvested and lysed using a Mini-BeadBeater (Biospec Products). DNA was sheared by ultrasonication using an immersion tip (Sonics VC-505). Cleared cell extracts were immunoprecipitated using EZview Red Anti-c-Myc Affinity Gel (Sigma-Aldrich Cat# E6654). Crosslinks were reversed in the eluted immunoprecipitated material by heating to 65˚C for at least 6 hours and the resulting DNA was purified and ethanol precipitated. Sequencing was done at the Genomic Sequencing and Analysis Facility of The University of Texas at Austin on Illumina Novaseq in 2 x 150 PE mode.

## CUT&Tag

CUT&Tag was performed as described previously [40,82] using the CUT&Tag-IT Assay Kit from Active Motif (Catalog No. 53160) and following manufacturer's instructions. Sequencing was done at the Genomic Sequencing and Analysis Facility of The University of Texas at Austin or the MD Anderson Cancer Center-Science Park NGS Facility using Illumina Novaseq in 2 x 150 or 2 x 75 PE mode.

## Cell cycle synchronization and single generation plasmid segregation

Cells were arrested in G1/S with α-factor, and released into the cell cycle by washing off the pheromone. Segregation of a fluorescence-tagged reporter plasmid was assayed in late anaphase cells with well separated mother and daughter nuclei.

## Plasmid stability under non-selective growth

Plasmid loss rate in the absence of selection was assayed as described previously [83]. The fractions of plasmid-containing cells in the population at the start ($f_0$) and at the end ($f_n$) of the growth period without selection for the plasmid-borne marker (n = 10 generations) were estimated by plating equal aliquots of the cultures on selective and non-selective plates. The plasmid loss rate per generation was estimated as $(1/n) \times \ln(f_0/f_n)$. The reporter plasmid used for this assay shown in S4A Fig contained the insertion of *ADE2* as the only modification to the 2-micron plasmid [35]. The stability of this plasmid approaches that of the native plasmid in a wild type [$Cir^0$] host strain. The reporter plasmid for the assay shown in Fig 12C [52], which is a yeast-*E. coli* shuttle vector, has a lower intrinsic stability when supplied with Rep1 and Rep2.

## Two-hybrid analysis

Protein interactions *in vivo* in yeast were assayed using the 'Matchmaker Gold Two-hybrid System' (Clontech) using the protocols provided by the supplier. Diploids formed by mating the 'bait'- and 'prey'-containing strains were spotted by serial dilution on appropriate selection media to test for the expression of reporter genes.

## Protein-protein interaction assays using affinity enrichment

Proteins of interest containing epitope tags or in-frame GST-fusions were expressed in yeast or in *E. coli*. The *RSC1* or *RSC2* locus in yeast was modified to express either of the tagged proteins from the native promoter. Expression cassettes for the galactose-inducible tagged Rep1 and Rep2 proteins were integrated into the chromosome. Tac-promoter-based expression systems and IPTG induction were utilized for protein expression in *E. coli*. Targeted proteins were pulled down from cell extracts using tag-specific affinity matrices [84,85]. IgG Sepharose 6 Fast Flow and Calmodulin Sepharose 4B were obtained from Cytiva, and 'EZView Red' Anti-HA and Anti-cMyc affinity gels from Sigma-Aldrich. The presence of candidate

interacting proteins trapped on the matrix was probed by western blotting or by mass spectrometry. Antibodies used in western blot assays were purchased from Covance, MilliporeSigma or Abcam. For mass spectrometry, RSC1 or RSC2 complex was enriched from yeast extracts using a TAP-tag based purification protocol [86]. The affinity-enriched protein fractions were concentrated by TCA precipitation, digested with trypsin, and processed in the UT Austin Biological Mass Spectrometry core facility using the DIonex LC and Orbitrap Fusion 2 instruments (LC-MS/MS). The raw datasets were analyzed using FragPipe-v17.1 software against the reference *S. cerevisiae* reference proteome.

### Preparation of chromosome spreads and probing by indirect immunofluorescence

Chromosome spreading was performed as described previously [87]. The slides carrying the spreads were treated with appropriate primary and secondary antibody combinations prior to fluorescence microscopy. The primary antibodies, mouse anti-Myc (1:300), rat anti-HA (3F10; 1:300), and mouse anti-GFP (1:300), were purchased from Roche (Germany). AlexaFluor488-conjugated goat anti-mouse antibody (1:200) and TRITC-conjugated goat anti-rat antibody (1:200) were obtained from Jackson ImmunoResearch Laboratories.

### Fluorescence microscopy

Fluorescence microscopy was performed in mildly fixed cells [37] or in chromosome spreads [88] using a Zeiss Axio Observer Z1 fluorescence microscope and ZEN 3.1 (Blue edition) software. Images acquired using the Z-stack mode at 0.20 μm intervals were opened in Imaris software (Bitplane, Imaris 8.0.2), and 'slice tool' was used to measure the spacing between the centroids of two fluorescent foci. In cells labeled with binary fluorescence (protein-protein or protein-plasmid), the degree of overlap between the two types of fluorescence was estimated as their Pearson's correlation coefficient (PCC) [89]. The threshold for each fluorescence emission channel was set using the 'automatic thresholding option' in the 'Coloc' tool of Imaris. The PCC values were determined for each individual cell image. Within the -1.0 to 1.0 range of PCC, 0.5 to 1.0 was taken to denote significant fluorescence overlap [90]. The validity of each estimated PCC was checked after randomizing the fluorescence overlap by rotating the pair of images for each cell through 90˚ relative to each other and recalculating PCC [91].

### Analysis of genomic data

ChIP-seq and CUT&Tag reads were aligned to a custom *S. cerevisiae* reference genome that included 2-micron plasmid sequences using bowtie2 (version 1.2.3) [92] with default parameters for ChIP-seq and options '—end-to-end—very-sensitive—no-unal—no-mixed—no-discordant—phred33 -I 10 -X 700' for CUT&Tag [82]. Peaks in ChIP-seq and CUT&Tag-seq data were identified using MACS2 (version 2.1.4) [93,94]. Visualization of enriched peaks in genome browser tracks, heatmaps and average profile plots (Figs 1–6) was done using the ppois bigwig tracks output by MACS2 and the Python packages pyGenomeTracks (version 3.7) [95,96] and deepTools (version 3.5.1) [97].

### Biological replicates and statistical methods

The quantitative data shown in Figs 10–12 and S12–S15 were obtained from three biological replicates. The N values given in histogram plots are the total number of cells scored from the combined replicates of individual assays. Data analyzed using GraphPad Prism software (8.4.3) are presented as mean ± SE. Statistical significance was estimated using Student's t-test.

## Supporting information

**S1 Table. Yeast strains and their relevant features.** The genotypes of the yeast strains used in the present study along with the figures depicting the experimental results obtained with them are listed. Strains containing or lacking the native 2-micron plasmid are indicated as [Cir$^+$] or [Cir$^0$], respectively.
(DOCX)

**S2 Table. Plasmids used in this study.** The plasmids listed below as groups I-III were used for yeast two-hybrid analyses, expression in *E. coli* and interaction assays, and cell biological/fluorescence microscopy experiments in yeast, respectively. The figures containing experimental results to which they contributed are indicated. For previously described plasmids, the relevant references are given.
(DOCX)

**S1 Fig. (accompanies Fig 1). Overview of Rep protein localization.** Binding of Rep1 and Rep2 assayed by ChIP-seq is shown using input-corrected signal tracks along with negative controls (No tag and Flp). The locations of snoRNAs, tRNAs and *CEN* are shown below each track using the colors indicated. **(A)** chrIII. **(B)** A region of chrI.
(TIF)

**S2 Fig. (accompanies Fig 1). Comparison of Galactose-induced versus natively expressed Rep proteins. (A)** Western blot analysis of strains expressing epitope-tagged Rep1 or Rep2 under the control of their native promoters on the 2-micron plasmid or the *GAL1-10* promoter. Gal and Glu refer to growth in medium with galactose or glucose as the carbon source. Blots were probed with anti-Myc (to visualize Rep1/Rep2) and β-actin (loading control) antibodies. **(B)** ChIP-PCR analysis of Rep1 and Rep2 localization at the *tV(UAC)D* locus (tRNA valine). WCE refers to the whole-cell extract, or input sample before ChIP, while ChIP refers to the ChIP sample with anti-Myc. PCR was performed for the tRNA valine locus and for a negative control locus (*TRP1*).
(TIF)

**S3 Fig. (accompanies Fig 3). Rep protein and Rsc2 localization at tRNA loci.** The top part of each panel shows the average binding profile and the bottom shows binding to each region as a heat map.
(TIF)

**S4 Fig. (accompanies Fig 4). (A)** Plasmid loss rate per generation, measured as described previously [83], is increased in both *rsc2Δ* and truncated *rsc2* (*rsc2 trn*). **(B)** Genome-wide correlation of binding as measured by the ChIP-seq assay for Rep1 and Rep2 in the indicated strain backgrounds. Rep1 shows better correlation genome-wide with Rep2 in the wild type (*wt*) strain than with Rep1 in the *rsc* mutant strains, and Rep2 behaves likewise. **(C-E)** Rep1 and Rep2 localization at tRNA and snoRNA loci in wild type, *rsc2Δ* and truncated *rsc2* strains as measured by ChIP-seq. In some instances, a redistribution of Rep1 and Rep2 can be seen around the original binding sites in the *rsc2* mutant strains.
(TIF)

**S5 Fig. (accompanies Fig 6). Rep2 localization at tRNAs, *TEL*s and *CEN*s under *rsc2* mutation.** Binding of Rep1 at **(A)** tRNA loci, **(B)** *TEL*s and **(C)** *CEN*s as measured by ChIP-seq.
(TIF)

**S6 Fig. (accompanies Fig 7). Rep1-Rep2 interaction in the two-hybrid assay. (A, B)** The interaction between Rep1 and Rep2 tested here served as a positive control for the assays

shown in Fig 7. This interaction was weaker with the [Rep2-BD]-[Rep1-AD] pair than with the [Rep1-BD]-[Rep2-AD] pair in both the [Cir$^0$] and [Cir$^+$] hosts. Presumably, the domain fusions in the former bait-prey configuration interferes partially with Rep1-Rep2 interaction.
(TIF)

**S7 Fig. (accompanies Fig 8). Rep1-Sfh1(Δ1–278) and Rep2-Sfh1(Δ1–278) interactions assayed using *E. coli*-expressed proteins. (A, B)** The assays were similar to those shown in Fig 8, except that Sfh1(Δ1–278) was expressed instead of Sfh1.
(TIF)

**S8 Fig. (accompanies Fig 8). Interactions of Sfh1 or Sfh1(Δ1–278) with Rep1 or Rep2 when the Rep proteins are co-expressed. (A, B)** The analyses were performed using extracts from *E. coli* strains expressing either Sfh1 or Sfh1(Δ1–278) together with both Rep1 and Rep2. The methodologies were analogous to those employed for the assays shown in Figs 8 and S7.
(TIF)

**S9 Fig. (accompanies Fig 9). Verification of [Rep1-Rep2]-RSC2 complex interaction by individual enrichment of Rep1, Rep2 or Rsc2.** The schematic diagram depicting the potential interaction of the RSC2 complex with Rep1-Rep2 is modeled after the corresponding diagram in Fig 9. The Rsc2 protein derivative for this assay carried the CBP-tag without the accompanying dual Protein A-tag. The primary enrichment was performed using anti-HA (left panel), anti-Myc (middle panel) or anti-CBP (right panel) antibody. Western blotting was performed with each of these antibodies to test the co-enrichment of suspected partner proteins.
(TIF)

**S10 Fig. (accompanies Fig 9). Interactions of Rep1 or Rep2 with the affinity-enriched RSC1 complex.** The schematic diagram (top) for Rep1-Rep2 interaction with the RSC1 complex is a close replica of that in Fig 9, with Rsc2 replaced by Rsc1. The epitope tags fused to Rep1, Rep2 and Rsc1 are indicated. The Table below at the left lists the relevant mass spectrometry data for the RSC1 complex obtained by enrichment on IgG-sepharose beads. The association of Rep1 or Rep2 with the enriched complex was probed using anti-HA or anti-Myc antibodies (directed to Rep1 or to Rep2, respectively) (right panel).
(TIF)

**S11 Fig. (accompanies Fig 9). Verification of [Rep1-Rep2]-RSC1 complex interaction by individual enrichment of Rep1, Rep2 or Rsc1.** The schematic diagram from S9 Fig is redrawn here with Rsc1-CBP replacing Rsc2-CBP. Enrichment of individual proteins and probing for associated proteins by western blotting were carried out as in the assays depicted in S9 Fig.
(TIF)

**S12 Fig. (accompanies Fig 10). Overlap between Rsc1 or Rsc2 fluorescence and Rep1 or Rep2 fluorescence in [Cir$^+$] chromosome spreads relative to fluorescence overlap by chance. (A, B)** The experimental protocols were as described under Fig 10. To obtain the degree of random overlap, PCC values were obtained for each spread after rotating the red and green fluorescence images through 90˚ relative to each other. Note that the median PCC values for experimentally observed (E) and randomized (R) fluorescence overlaps in all three plots in **(B)** are below 0.3, the lower threshold set for 'partial overlap' (see Fig 10).
(TIF)

**S13 Fig. (accompanies Fig 10). Overlap between Rsc1 and Rsc2 in mitotic chromosome spreads.** The epitope-tagged Rsc1 and Rsc2 were visualized by immunofluorescence in [Cir$^+$] mitotic spreads. The extent of the experimentally observed Rsc1-Rsc2 fluorescence overlap (E)

and the overlap following randomization (R) are plotted (see Fig 10). Bar = 5 μm.
(TIF)

**S14 Fig. (accompanies Fig 10). Rsc1 or Rsc2 fluorescence overlap with Rep1 or Rep2 fluorescence in [Cir⁰] chromosome spreads relative to fluorescence overlap by chance. (A, B)**
The analysis was done as described in the legend to S12 Fig, except that [Cir$^0$] chromosome spreads were assayed. The median PCC values for fluorescence overlap in the experimental (E) and randomized (R) samples in all the plots are < 0.3, signifying no overlap (see Fig 10).
(TIF)

**S15 Fig. (accompanies Fig 10). Localization of Rep1 and Rep2 with respect to Rsc2 in meiotic chromosome spreads.** Chromosome spreads were prepared from [Cir$^0$] diploid cells transferred to sporulation medium for 8 hr and were assayed by immunofluorescence microscopy. The native *RSC2* locus was modified to express Rsc2-HA. Rep1-Myc or Rep2-Myc, expressed under *GAL* promoter control from an integrated cassette, was complemented by its untagged Rep partner expressed by a *CEN*-plasmid from the *GAL* promoter. A β-estradiol inducible activator system [98] was used to control the *GAL* promoter. Bar = 5 μm.
(TIF)

**S16 Fig. (accompanies Fig 11). A nearly single-copy yeast plasmid whose partitioning capability can be modified by manipulating the host strain and the carbon source.** The features of the pSG1 plasmid are schematically diagrammed. In addition to the *TRP1* marker for selection in yeast and [LacO]$_{256}$, the plasmid carries the *ORI-STB* sequence from the 2-micron plasmid and a *CEN* sequence whose function is regulated by the *GAL* promoter [33]. The plasmid behaves as pSG1-*CEN* in a [Cir$^0$] host grown in glucose (Rep1-Rep2 proteins absent; *CEN* active) and as pSG1-*ARS* when this strain is shifted to galactose (*CEN* inactive). In a [Cir$^0$] host expressing Rep1 and Rep2 under *GAL* promoter control, the plasmid is pSG1-*STB* in the presence of galactose.
(TIF)

## Acknowledgments

Illumina sequencing was performed by the Genomic Sequencing and Analysis Facility at UT Austin, Center for Biomedical Research Support (RRID: SCR_021713) and by the NGS Facility at the MD Anderson Cancer Center, Science Park (which was supported by CPRIT Core Facility Support Grant RP170002). Mass spectrometry and protein identification was performed by the UT Austin Center for Biomedical Research Support Biological Mass Spectrometry Facility (RRID: SCR_021728). Computational analyses were performed using the Biomedical Research Computing Facility at UT Austin, Center for Biomedical Research Support. (RRID#: SCR_021979) and the Texas Advanced Computing Center (TACC). We thank Ruth Raichur, Ethan Chen, Shi En Lin and Neeti Patel for technical assistance and Trevor Freeman and Muyoung Lee for assistance with initial analysis of the ChIP-seq data.

## Author Contributions

**Conceptualization:** Makkuni Jayaram, Vishwanath R. Iyer.

**Data curation:** Chien-Hui Ma, Deepanshu Kumar, Makkuni Jayaram, Vishwanath R. Iyer.

**Formal analysis:** Vishwanath R. Iyer.

**Funding acquisition:** Makkuni Jayaram, Vishwanath R. Iyer.

**Investigation:** Chien-Hui Ma, Deepanshu Kumar, Vishwanath R. Iyer.

**Project administration:** Makkuni Jayaram, Vishwanath R. Iyer.

**Supervision:** Makkuni Jayaram, Santanu K. Ghosh, Vishwanath R. Iyer.

**Validation:** Chien-Hui Ma, Deepanshu Kumar, Makkuni Jayaram, Santanu K. Ghosh.

**Visualization:** Deepanshu Kumar, Vishwanath R. Iyer.

**Writing – original draft:** Makkuni Jayaram, Santanu K. Ghosh, Vishwanath R. Iyer.

**Writing – review & editing:** Chien-Hui Ma, Deepanshu Kumar, Makkuni Jayaram, Santanu K. Ghosh, Vishwanath R. Iyer.

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
