## [Decision Letter · Decision Letter 0]

25 Jul 2023

Dear Dr Jayaram,

Thank you very much for submitting your Research Article entitled 'The selfish yeast plasmid exploits a SWI/SNF-type chromatin remodeling complex for hitchhiking on chromosomes and ensuring high-fidelity propagation' to PLOS Genetics.

The manuscript was fully evaluated at the editorial level and by independent peer reviewers. The reviewers appreciated the attention to an important problem, but raised some substantial concerns about the current manuscript. Based on the reviews, we will not be able to accept this version of the manuscript, but we would be willing to review a much-revised version. We cannot, of course, promise publication at that time.

Overall, the reviewers raised several points related to experimental design and data quality, interpretation and visualization. In addition, it was suggested to test the tethering assay with non-RSC proteins; this seems a useful suggestion to further dissect whether the rescue of segregation is RSC-specific. One reviewer questioned the novelty of the findings -- we ask you to clearly highlight which results are novel and which are confirmatory. Please make sure that all large datasets are made available through a public repository.

If you decide to revise the manuscript for further consideration at PLOS Genetics, please aim to resubmit within the next 60 days, unless it will take extra time to address the concerns of the reviewers, in which case we would appreciate an expected resubmission date by email to plosgenetics@plos.org.

We are sorry that we cannot be more positive about your manuscript at this stage. Please do not hesitate to contact us if you have any concerns or questions.

Yours sincerely,

Bas van Steensel

Academic Editor

PLOS Genetics

Eva Stukenbrock

Section Editor

PLOS Genetics

Reviewer's Responses to Questions

**Comments to the Authors:**

Reviewer #1: The manuscript by Ma et al. identifies a direct role for the RSC2 chromatin remodeling complex in persistence of the yeast 2 micron plasmid, a genetic element that segregates with chromosome-like efficiency. The Jayaram lab has laid out the “hitch-hiking” hypothesis in which plasmid-encoded proteins (Rep1 and Rep2) confer stability by tethering replicated plasmids to segregating sister chromatids. In this work, the authors map locations of Rep1/2 to chromosomes by ChIP-Seq and CUT&Tag and find overlap with the binding sites for RSC2, a chromatin remodeling complex previously implicated in 2 micron persistence. Enrichment sites include centromeres, telomeres and pol III transcribed genes, and binding to these sites is altered by loss of RSC2. There is significant variability in the strength of signal relative to controls but the general picture is sufficiently convincing. The authors go further by to show with mass spec, reciprocal IP, two-hybrid and colocalization that the Rep1 interacts with a subunit of RSC2, suggesting that it is the chromatin remodeler and not remodeled chromatin alone that facilitates recruitment. The authors use cytology in an attempt to show that segregating plasmids are enriched near a representative tRNA. Finally, they provide proof-of-principle of the hitchhiking hypothesis by showing that segregation can be restored by tethering a plasmid without a segregation system to RSC2. I have expressed some concerns in the comments below. Otherwise, I believe this work adds significant mechanistic understanding of how an extrachromosomal selfish genetic element persists in yeast.

Comments.

1) A reasonable concern is that the association of Rep1/2 to RSC2 chromosomal binding sites (Figs 1-6, 8 as well as S9-S10) is an artifact of overexpression. The authors should acknowledge this caveat but then justify to the reader why the data is still admissible. Are there any experiments are is it merely the bulk of consistent data?

2) The authors provide representative examples of Rep1/2 mapping in figure 1 (presumably the most convincing examples) and then follow with heat maps in subsequent figures. The data are initially described as providing “clear localization”/”consistently high signals” when the data are actually less than perfect. I believe it is a disservice to the reader to 1) not acknowledge the limitations up front and that 2) the data must be viewed in aggregate to draw conclusions. The authors eventually point to the substantial signal for the Flp control by CUT&Tag at CEN5 and TEL15 in the examples of figure 1. I believe it is more helpful to make this point with the heat maps where it is shown how different techniques support their conclusion better at different sites. Specifically in figure 2, the full magnitude of the Flp control signal by CUT&Tag can be appreciated but counterbalanced by the ChIP-Seq data. Additionally, figure 2 shows that the No Ab control for CUT&Tag for telomeres has the highest signal of all. This should be acknowledged.

3) While the bulk of tRNAs may localize to the CEN cluster or nucleolus, some tRNAs do not localize to either (Belagal Gadal 16; Chen Gartenberg 14). The localization of the tRNA in the colocalization study of figure 9 must be known to draw the conclusions the authors state. When the plasmid is near the tRNA, is the tRNA near CENs or the nucleolus. This would be reasonable to test or the interpretation should be modified. Additionally, some of the measurements are made in glucose while others are made in galactose, which induces changes in genome organization (Brickner lab and others) that could affect the results. In addition, why choose this tRNA gene? Did it have a strong mapping signal?

4) Figure 10 provides a proof-of-principle that tethering a plasmid to RSC2 confers segregation. My concern is that similar results might be obtained by tethering the plasmid to any chromosome bound protein. If so, then this experiment tells us that hitchhiking can occur but does not inform on whether RSC normally carries out this function. Are there classes of proteins for which this would work and those for which it would not?

Minor comments

1) While pericentric chromatin may be unusual in yeast, I am not aware of it satisfying heterochromatic criteria. Do the Sir proteins silence genes there?

2) Figures S1 and S2 were less ambiguous to me than the current Figure 1. The authors must have some reason for delegating them to the supplement.

3) In summarizing the initial mapping data, the authors suggest that plasmids might not be tethered to all sites because there are only a small number of plasmid foci in dividing cells. I am not certain I understand the argument. CENs, TELs and tRNAs all cluster so might the plasmids be localized to these clusters, thus yielding occupancy at all sites? Or is it that only a subset of the hundreds of sites are occupied in a single cell but that the population average makes it appear as if the 100s of sites are always occupied?

3) Does the Rsc2-truncation mutant retain some remodeling activity? If not, does the residual signal of Rep1 binding in this mutant (Fig 4) indicate that recruitment is independent of remodeling? I think that would be a significant claim.

4) Either there is a typo in the y-axis label of figure 10A (the mis-segregation panel) or I do not understand the description in the figure legend.

Reviewer #2: Ma et al. explore the genomic location of the 2-micron plasmid partitioning proteins Rep1 and Rep2 by two different techniques (ChIP seq and CUT&Tag) and show that they overlap with the RSC2 remodeling complex and that their location moderately depends on RSC2. They then explore the interaction between Rep1 and Rep2 with RSC components by two hybrid assays, pull down and colocalization experiments and show that tethering a plasmid without the partitioning system to RSC can confer the plasmid the ability to equally segregate during host cell division. Although the authors provide new data, the main observations presented in the manuscript are already published. [Rep1-Rep2]-RSC interaction is shown in Ma et al., Nucleic Acids Research, 2013 (reference 35 in the manuscript) and the requirement of RSC2 for Rep1 localization is shown in Wong et al., Mol Cell Biol, 2002 (reference 36 in the manuscript). Therefore, in my opinion the novelty of the research is not enough for publication in PLOS Genetics. The authors could use their data to explore new questions about the 2-micron plasmid partitioning system (for example, search for other Rep1-Rep2 tethering factors in chromatin additional to RSC2 or understand why the interaction [Rep1-Rep2]-RSC1 is not functionally relevant) rather than confirm previous observations with new techniques. Moreover, several important aspects of the manuscript must be improved:

- The figures are not well organized. For example, the authors use three main figures to depicture the genomic location of Rap1 and Rap2 and three additional figures to show how it changes in rsc2Δ and Rsc2 truncation. These six figures could be condensed in one. Also, all the pull-down experiments are in supplementary figures while just a two-hybrid assay is using a main figure.

- Rep1 and Rep2 genomic location obtained by from ChIP seq and CUT&Tag show discrepancies and, in some cases, the signal is not above background (especially in the case of Rap2). It can be informative to subtract the background and examine which peaks overlap in the two techniques.

- The differences in Rap1 and Rap2 localization in rsc2Δ and Rsc2 truncation should be confirmed by a quantitative technique such as ChIP-qPCR.

- Mass spectrometry data should be made available in an open access repository.

- I consider than the most interesting part of the manuscript is the artificially binding of an ARS plasmid to Sfh1 to achieve equal segregation. The authors could use this system to search for additional Rep1-Rep2 interaction partners with relevance in plasmid segregation.

**Have all data underlying the figures and results presented in the manuscript been provided?**

Reviewer #1: Yes

Reviewer #2: **No: **Mass spectrometry data is not made available in public repositories

PLOS authors have the option to publish the peer review history of their article (what does this mean?). If published, this will include your full peer review and any attached files.

Reviewer #1: No

Reviewer #2: No

---

## [Decision Letter · Decision Letter 1]

19 Sep 2023

Dear Dr Iyer,

We are pleased to inform you that your manuscript entitled "The selfish yeast plasmid exploits a SWI/SNF-type chromatin remodeling complex for hitchhiking on chromosomes and ensuring high-fidelity propagation" has been editorially accepted for publication in PLOS Genetics. Congratulations!

We recommend, however, at your discretion, that you look once more look at the remaining comments and consider to make a few minor  changes that could help to further clarify and balance the manuscript - in particular the four remaining points of Reviewer #1. 

Regarding point 3 of this reviewer (and a point previously raised by reviewer #2): we agree that without repeating the tethering experiment with a set of other chromosomal proteins, it remains unclear whether the results of this assay are *specific* for the RSC complex. We strongly recommend that this limitation is explicitly discussed, either at the end of the relevant Results section, or in the Discussion section. 

Reviewer #2 still appears unconvinced of the novelty of some of the work; as other readers may also wonder about this, it could help to explicitly highlight the differences between the current work and the  Wong and Ma papers (refs 35 and 36) in a brief Discussion paragraph.

Finally, for the sake of transparency, it would be good to include the ChIP-seq correlation heatmap that you depicted in your rebuttal letter as a supplementary figure (again, so that other readers may also appreciate this analysis).

Yours sincerely,

Bas van Steensel

Academic Editor

PLOS Genetics

Eva Stukenbrock

Section Editor

PLOS Genetics

Comments from the reviewers (if applicable):

Reviewer's Responses to Questions

**Comments to the Authors:**

Reviewer #1: The authors have addressed most of my concerns.

1) I am convinced with the aggregate of the ChIP-Seq and Cut&Tag data that Rep1 and Rep2 colocalize with the RSC2 complex across the genome, even if there are exceptions and high Flp background in a number of instances. For example, in fig 1 the FLP values for CUT&Tag are nearly as high as Rep2 for tRNAval, CEN5 and the TELs. Thus, I think the term "clear localization" on bottom of page 7 is used prematurely before the reader has been told that the data is better considered in aggregate.

2) The authors misrepresent the work of Sharp et al., 2002, Genes & Dev 16: 85-100, which has nothing to do with silencing at centromeres.

3) I like the experiment in figure 12. I think it shows that tethering a plasmid to a protein that binds chromosomes confers segregation properties to that plasmid. At best you could say that tethering the plasmid to sites where RSC binds confers segregation properties to that plasmid. As I said before, tethering that plasmid to histone might do the same thing. I think the authors ought to be forthright about what this experiment tells us. Adding the word “bypass” to the section title is not enough. You could say the data are consistent with the model of RSC being a docking site for plasmids but…. What this experiment needs is some evidence that not all tethers work. For example, what about tethering the plasmid to other remodelers. Perhaps, only certain types of remodeling are sufficient. Check out Munoz Uhlmann 2019 figure 6 to see how this type of tethering experiment can be profoundly informative.

4) Fig1 conveniently labels the techniques that were used for each panel ChIP-Seq vs Cut&Tag). This labeling is missing from subsequent figs, thereby forcing reader to look at legend. Add these labels.

Reviewer #2: I consider that even though genome wide mapping of Rep1/Rep2 is an advance in knowledge, it does not reveal any mechanism of action of these partitioning proteins as the authors claim.

In addition, the authors haven't address any of my concerns about figure distribution or controls such as qPCR (that could be done for the peaks that the authors show in the main figure).

**Have all data underlying the figures and results presented in the manuscript been provided?**

Reviewer #1: Yes

Reviewer #2: Yes

PLOS authors have the option to publish the peer review history of their article (what does this mean?). If published, this will include your full peer review and any attached files.

Reviewer #1: No

Reviewer #2: No

**Data Deposition**

http://datadryad.org/submit?journalID=pgenetics&manu=PGENETICS-D-23-00727R1

**Press Queries**

---

## [Editor Report · Acceptance letter]

4 Oct 2023

PGENETICS-D-23-00727R1 

The selfish yeast plasmid exploits a SWI/SNF-type chromatin remodeling complex for hitchhiking on chromosomes and ensuring high-fidelity propagation 

Dear Dr Iyer, 

We are pleased to inform you that your manuscript entitled "The selfish yeast plasmid exploits a SWI/SNF-type chromatin remodeling complex for hitchhiking on chromosomes and ensuring high-fidelity propagation" has been formally accepted for publication in PLOS Genetics! Your manuscript is now with our production department and you will be notified of the publication date in due course.

With kind regards,

Timea Kemeri-Szekernyes

PLOS Genetics

On behalf of:
